# GC content shapes mRNA storage and decay in human cells

**Maïté Courel[1], Yves Clément[2], Clémentine Bossevain[1], Dominika Foretek[3], Olivia Vidal Cruchez[4], Zhou Yi[5], Marianne Bénard[1], Marie-Noëlle Benassy[1], Michel Kress[1], Caroline Vindry[6], Michèle Ernoult-Lange[1], Christophe Antoniewski[7], Antonin Morillon[3], Patrick Brest[4], Arnaud Hubstenberger[5], Hugues Roest Crollius[2], Nancy Standart[6], Dominique Weil[1]***

[1]Sorbonne Université, CNRS, Institut de Biologie Paris Seine (IBPS), Laboratoire de Biologie du Développement, Paris, France; [2]Ecole Normale Supérieure, Institut de Biologie de l'ENS, IBENS, Paris, France; [3]ncRNA, Epigenetic and Genome Fluidity, Institut Curie, PSL Research University, CNRS UMR 3244, Sorbonne Université, Paris, France; [4]Université Côte d'Azur, CNRS, INSERM, IRCAN, FHU-OncoAge, Nice, France; [5]Université Côte d'Azur, CNRS, INSERM, iBV, Nice, France; [6]Department of Biochemistry, University of Cambridge, Cambridge, United Kingdom; [7]Sorbonne Université, CNRS, Institut de Biologie Paris Seine (IBPS), ARTbio Bioinformatics Analysis Facility, Paris, France

**Abstract** mRNA translation and decay appear often intimately linked although the rules of this interplay are poorly understood. In this study, we combined our recent P-body transcriptome with transcriptomes obtained following silencing of broadly acting mRNA decay and repression factors, and with available CLIP and related data. This revealed the central role of GC content in mRNA fate, in terms of P-body localization, mRNA translation and mRNA stability: P-bodies contain mostly AU-rich mRNAs, which have a particular codon usage associated with a low protein yield; AU-rich and GC-rich transcripts tend to follow distinct decay pathways; and the targets of sequence-specific RBPs and miRNAs are also biased in terms of GC content. Altogether, these results suggest an integrated view of post-transcriptional control in human cells where most translation regulation is dedicated to inefficiently translated AU-rich mRNAs, whereas control at the level of 5' decay applies to optimally translated GC-rich mRNAs.

*For correspondence: dominique.weil@upmc.fr

**Competing interests:** The authors declare that no competing interests exist.

## Introduction

Translation, storage, localization and decay of mRNAs in the cytoplasm are closely coupled processes, which are governed by a large number of RNA-binding proteins (RBPs) (*Hentze et al., 2018*). These RBPs have to act in a coordinated manner to give rise to a proteome both coherent with cellular physiology and responsive to new cellular needs. mRNA fate is also intimately linked with their localization in membrane-less organelles, such as P-bodies (PBs). We recently identified the transcriptome and proteome of PBs purified from human cells. Their analysis showed that human PBs are broadly involved in mRNA storage rather than decay (*Hubstenberger et al., 2017*; *Standart and Weil, 2018*), as also observed using fluorescent decay reporters (*Horvathova et al., 2017*). However, the mechanism underlying the large but specific targeting of mRNAs to PBs is still unknown, though it clearly results in the co-recruitment of particular RBPs (*Hubstenberger et al., 2017*).

In mammalian cells, the RNA helicase DDX6, known for its involvement in mRNA decay and translation repression, is a key factor in PB assembly (*Minshall et al., 2009*). Patients with neurodevelopmental delay caused by heterozygous DDX6 missense mutations were recently identified, and their skin fibroblasts show a PB defect (*Balak et al., 2019*). Human DDX6 interacts with both translational repressors, and the decapping enzyme DCP1/2 and its activators (*Ayache et al., 2015*; *Bish et al., 2015*). Its yeast homologue Dhh1 is a cofactor of DCP2, as well as a translational repressor (*Coller and Parker, 2005*). The RBP PAT1B has also been defined as an enhancer of decapping, as it interacts with DDX6, the LSM1-7 heptamer ring and the decapping complex in mammalian cells (*Vindry et al., 2017*), while in yeast Pat1p activates Dcp2 directly (*Nissan et al., 2010*) and its deletion results in deadenylated but capped intact mRNA (*Bonnerot et al., 2000*; *Bouveret et al., 2000*). DDX6 and PAT1B interact with the CCR4-NOT deadenylase complex and the DDX6-CNOT1 interaction is required for miRNA silencing (*Vindry et al., 2017*; *Chen et al., 2014*; *Mathys et al., 2014*; *Ozgur et al., 2015*). DDX6 also binds the RBP 4E-T, another key factor in PB assembly, which in turn interacts with the cap-binding factor eIF4E and inhibits translation initiation, including that of miRNA target mRNAs (*Kamenska et al., 2016*). Altogether, DDX6 and PAT1B have been proposed to link deadenylation/translational repression with decapping. Finally, the 5′−3′ exonuclease XRN1 decays RNAs following decapping by DCP1/2, a step triggered by deadenylation mediated by PAN2/3 and CCR4-NOT or by exosome activity (*Łabno et al., 2016*).

A number of RBPs also control mRNA fate in a sequence-specific manner, some of them localizing in PBs as well. For instance, the CPEB complex, best described in *Xenopus* oocytes (*Minshall et al., 2007*), binds the CPE motif in the 3′ untranslated region (UTR) of maternal transcripts through CPEB1, thus controlling their storage and their translational activation upon hormone stimulation (*Standart and Minshall, 2008*). Additional examples include the proteins which bind 3′UTR AU-rich elements (ARE), such as HuR and TTP, to control translation and decay, and play key roles in inflammation, apoptosis and cancer (*Wells et al., 2017*). Protein-binding motifs are generally not unique and rather defined as consensus sequence elements. In the case of RISC, binding specificity is given by a guide miRNA, which also hybridizes with some flexibility with complementary mRNA sequences. A variety of techniques have therefore been developed to identify the effective RNA targets of such factors, ranging from affinity purification (such as RIP or CLIP) to transcriptome and polysome profiling after RBP silencing, providing the groundwork to address systematic questions about post-transcriptional regulation.

In this study, we searched for broad determinants of mRNA storage and decay in unstressed human cell lines, using our transcriptome of purified PBs and several transcriptomic analyses performed after silencing of general translation and decay regulatory factors, including DDX6, PAT1B and XRN1. We also used datasets available from the literature, including a transcriptomic analysis after DDX6 silencing, a DDX6-CLIP experiment and various lists of RBP and miRNA targets. Their combined analysis revealed the central role of mRNA GC content which, by impacting codon usage, PB targeting and RBP binding, influences mRNA fate and contributes to the coordination between two opposite processes: decay and storage. Reporter mRNAs varying in their GC content confirmed that AU-rich mRNAs have a lower protein yield than GC-rich ones, that they preferentially localize to PBs, and that they have an enhanced capacity to form RNP granules in vitro.

## Results

### PBs mostly accumulate AU-rich mRNAs

We have previously shown that PBs store one third of the coding transcriptome in human epithelial HEK293 cells (*Hubstenberger et al., 2017*). Such a large transcript number led us to search for general distinctive sequence features that could be involved in PB targeting. We first analyzed transcript length, as it was reported to be key for mRNA accumulation in stress granules (*Khong et al., 2017*). When mRNAs were subdivided into six classes ranging from <1.5 kb to >10 kb, longer mRNAs appeared more enriched in PBs than shorter ones, with a moderate correlation between length and PB enrichment (Spearman r ($r_s$) = 0.39, p<0.0001) (*Figure 1A*, *Figure 1—figure supplement 1A,B*). However, their increased length in PBs was less striking than previously observed for stress granule mRNAs (*Khong et al., 2017*) (*Figure 1—figure supplement 1C*).

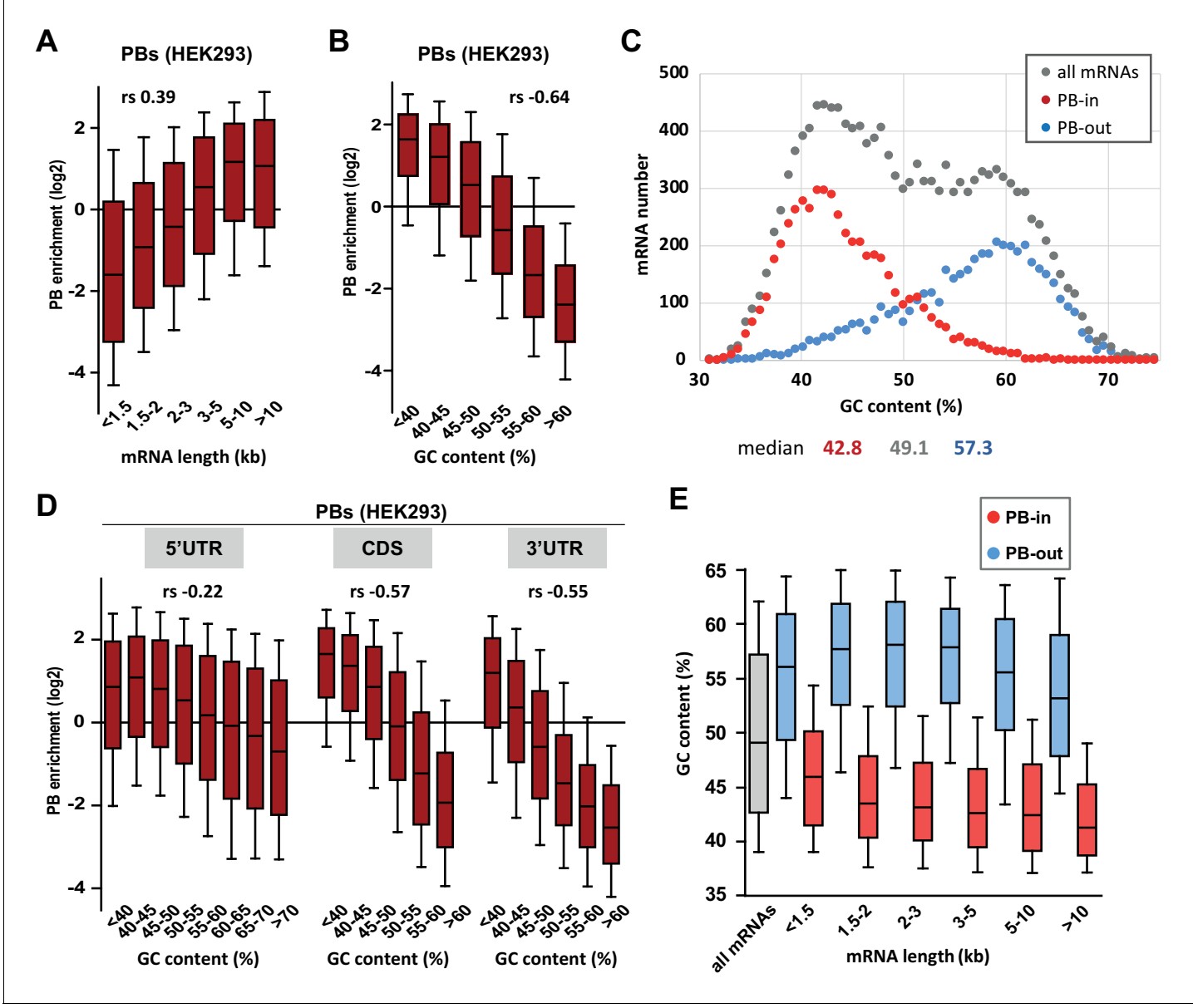

**Figure 1.** PB mRNAs are AU-rich and longer than average. (A) Long mRNAs are particularly enriched in PBs. Transcripts were subdivided into six classes depending on their length (from <1.5 kb to >10 kb). The boxplots represent the distribution of their respective enrichment in PBs. The boxes represent the 25–75 percentiles and the whiskers the 10–90 percentiles. rs, Spearman correlation coefficient. (B) AU-rich mRNAs are particularly enriched in PBs. Transcripts were subdivided into six classes depending on their GC content (from <40 to >60%) and analyzed as in (A). (C) PBs mostly contain the AU-rich fraction of the transcriptome. The human transcriptome was binned depending on its GC content (0.7% GC increments). The graph represents the number of PB-enriched (PB-in, p<0.05, n = 5200) and PB-excluded (PB-out, p<0.05, n = 4669) transcripts in each bin. The distribution of all transcripts is shown for comparison (n = 14443). The median GC value is indicated below for each group. (D) mRNA localization in PBs mostly depends on the GC content of their CDS and 3'UTR. The analysis was repeated as in (B) using the GC content of the 5'UTR, CDS or 3'UTR, as indicated. For 5'UTRs, the >60% class was subdivided into three classes to take into account their higher GC content compared to CDSs and 3'UTRs. −0.57 and −0.55 are not significantly different (p=0.17), while −0.22 and −0.55 are (p<0.0001) (E) GC content is lower in PB-enriched mRNAs than PB-excluded ones independently of their length. The GC content distribution of PB-enriched (PB-in, p<0.05) and PB-excluded (PB-out, p<0.05) mRNAs was analyzed as in (B).

The online version of this article includes the following figure supplement(s) for figure 1:

**Figure supplement 1.** PB-enriched mRNAs tend to be long and AU-rich.

Most remarkably, mRNA accumulation in PBs was dependent on their global nucleotide composition, with a strong correlation between GC content and PB localization ($r_s = -0.64$, p<0.0001). When transcripts were subdivided into six classes ranging from <40% to >60% GC, PB enrichment was predominant for those <45% GC (*Figure 1B*, *Figure 1—figure supplement 1D,E*). While reminiscent of the low GC content reported for stress granule mRNAs in HEK293 cells (*Khong et al., 2017*), our reanalysis of the published dataset indicated that stress granule localization correlated weakly with the gene GC content ($r_s = -0.12$, p<0.0001) and almost not at all with the mRNA GC content ($r_s = -0.06$, p<0.0001). Indeed, comparing the GC content distribution of the transcripts that are enriched or excluded from PBs with all HEK293 cell transcripts, revealed that mRNA storage in PBs is confined to the AU-rich fraction of the transcriptome (*Figure 1C*).

As these transcripts also correspond to AU-rich genes (*Figure 1—figure supplement 1F*), it raised the possibility that the impact of GC content on PB enrichment resulted indirectly from the genomic context of the genes. To address this issue, we looked at the link between PB enrichment and meiotic recombination, which can influence GC content through GC-biased gene conversion (*Duret and Galtier, 2009*). The correlation between PB enrichment and meiotic recombination was much weaker than between PB enrichment and mRNA GC content ($r_s = -0.16$ vs $-0.64$, p<0.0001 for both, significantly different from each other, p<0.0001). Moreover, the latter was almost unchanged when controlling for meiotic recombination ($r_s = -0.65$ vs $-0.64$, p<0.0001). Finally, it was still significant when controlling for intronic or flanking GC content ($r_s = -0.33$ and $-0.45$ respectively, all p<0.0001), showing that mRNA base composition and PB enrichment are associated independently of meiotic recombination or the genomic context. We also computed partial correlations to verify that the correlation between PB enrichment and GC content was not secondary to the correlation that exists between GC content and expression level, or between GC content and gene conservation (*Figure 1—figure supplement 1G*).

To refine the link between mRNA accumulation in PBs and their GC content, we analyzed separately the influence of their CDS and UTRs. Interestingly, mRNA accumulation in PBs correlated strongly with the GC content of both their CDS and 3'UTR ($r_s = -0.57$ and $-0.55$, respectively, p<0.0001 for both), and weakly with the one of their 5'UTR ($r_s = -0.22$, p<0.0001) (*Figure 1D*, *Figure 1—figure supplement 1E*). Moreover, the lower GC content of PB-enriched mRNAs compared to PB-excluded ones was a feature independent of their length, since it was observed in all length ranges (*Figure 1E*, *Figure 1—figure supplement 1B*). Conversely, the longer length of PB mRNAs was a feature independent of their GC content (*Figure 1—figure supplement 1E,H*).

In conclusion, while PB mRNAs tend to be longer than average, their most striking feature is that they correspond to an AU-rich subset of the transcriptome.

## GC bias in PBs impacts codon usage and protein yield

The strong GC bias in the CDS of PB mRNAs prompted us to compare the coding properties of PB-stored and PB-excluded mRNAs. Consistently, we found that the frequency of amino acids encoded by GC-rich codons (Ala, Gly, Pro) was lower in PB-stored than in PB-excluded mRNAs, while the frequency of those encoded by AU-rich codons (Lys, Asn) was higher (*Figure 2A*). The difference could be striking, as illustrated by Lys, whose median frequency in PB-excluded mRNAs was 32% lower than in PB-enriched mRNAs, thus ranging within the lower 17th centile of their distribution (*Figure 2—figure supplement 1A*). In addition to different amino acid usage, we observed dramatic variation in codon usage between the two mRNA subsets. For all amino acids encoded by synonymous codons, the relative codon usage in PBs versus out of PBs was systematically biased towards AU-rich codons (log2 of the ratio >0, *Figure 2B*). For example, among the six Leu codons, AAU was used 4-fold more frequently in PB-enriched than in PB-excluded mRNAs, whereas CUG was used 2-fold less frequently. This systematic trend also applied to Stop codons. Some additional codon bias independent of base composition (NNA/U or NNG/C) was also observed for 4 and 6-fold degenerated codons (*Figure 2—figure supplement 1B,C*). For instance, Leu was encoded twice more often by CUU than CUA in PB-enriched mRNAs, whereas the use of both codons was low in PB-excluded mRNAs. Similarly, Gly was encoded more often by GGG than GGC in PB mRNAs, whereas the use of both codons was similar in PB-excluded mRNAs (*Figure 2C*).

In human, 22 out of the 29 synonymous codons that are less frequently used (normalized relative usage <1) end with an A or U, and were therefore overrepresented in PB mRNAs (*Figure 2—figure supplement 1D*). Considering for each amino acid the codon with the lowest usage (called low

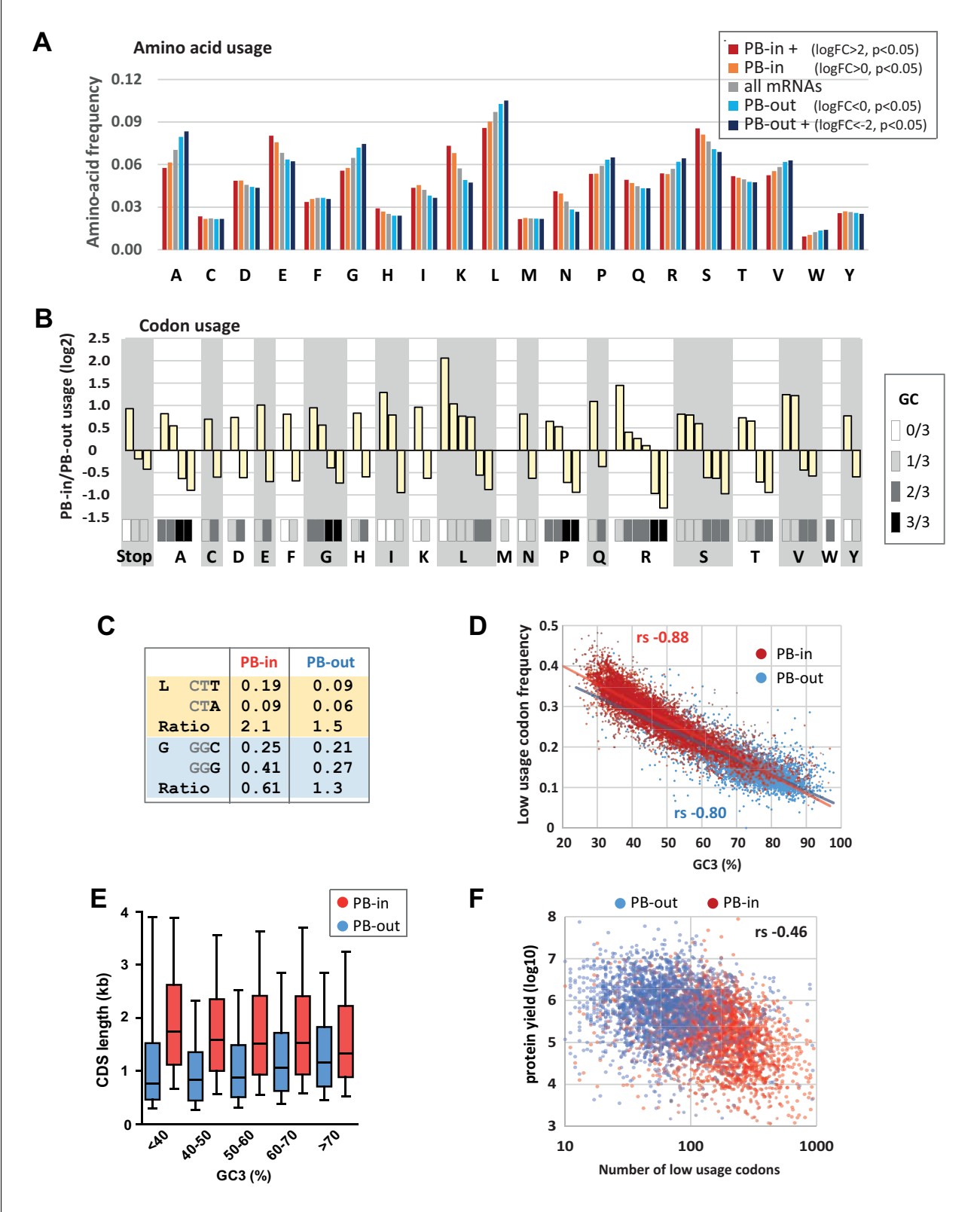

**Figure 2.** Codon usage is strongly biased in PBs. (**A**) PB mRNAs and PB-excluded mRNAs encode proteins with different amino acid usage. The graph represents the frequency of each amino acid in the proteins encoded by mRNAs enriched or excluded from PBs, using the indicated PB enrichment thresholds. (**B**) Codon usage bias in and out of PBs follows their GC content. The relative codon usage for each amino acid was calculated in PB-enriched (PB-in) and PB-excluded (PB-out) mRNAs, using a PB enrichment threshold of +/- 1 (in log2). The graph represents the log2 of their ratio (PB-

*Figure 2 continued on next page*

*Figure 2 continued*

in/PB-out) and was ranked by decreasing values for each amino acid. The GC content of each codon is gray-coded below, using the scale indicated on the right. (C) The usage of some codons is biased independently of their GC content. Two examples are shown encoding Leucine (L) and Glycine (G). (D) The frequency of low usage codons strongly correlates with the GC content of the CDS, independently of their PB localization. The frequency of low usage codons was calculated for mRNAs excluded (PB-out) and enriched (PB-in) in PBs using a PB enrichment threshold of +/- 1 (in log2). It was expressed as a function of the CDS GC content at position 3 (GC3). Note that the slopes of the tendency curves are similar for PB-enriched and PB-excluded transcripts. The difference between the Spearman correlation coefficients (rs) are nevertheless statistically significant (p<0.0001). (E) PB mRNAs have longer CDS than PB-excluded mRNAs. The analysis was performed as in *Figure 1E*. (F) The number of low usage codons per CDS is a good determinant of both protein yield and PB localization. The protein yield was expressed as a function of the number of low usage codons for PB-enriched (PB-in) and PB-excluded (PB-out) mRNAs. rs, Spearman correlation coefficient.

The online version of this article includes the following figure supplement(s) for figure 2:

**Figure supplement 1.** Amino acid usage and codon usage biases in PBs.

**Figure supplement 2.** Codon usage biases and abundance of amino-acylated tRNA.

usage codon thereafter), 14 out of 18 are NNA or NNU, with the exception of Thr, Ser, Pro, Ala. We calculated the frequency of low usage codons for each CDS, and plotted it as a function of the GC content at the third position (GC3) to avoid any confounding effects of the amino acid bias. As expected, the frequency of low usage codons correlated strongly and negatively with GC3, with AU-rich CDS having a higher frequency of low usage codons than GC-rich CDS (*Figure 2D*). According to their distinct GC content, PB mRNAs had a higher frequency of low usage codons than PB-excluded mRNAs. However, the correlation coefficient between frequency of low usage codons and GC3 was very close for both mRNA subsets ($r_s = -0.88$ for PB-enriched; $-0.80$ for PB-excluded mRNAs, p<0.0001 for both), meaning that their different frequency of low usage codons could be largely explained by their GC bias alone.

We previously reported that protein yield, defined as the ratio between protein and mRNA abundance in HEK293 cells, was 20-times lower for PB-enriched than PB-excluded mRNAs. This was not due to translational repression within PBs, as the proportion of a given mRNA in PBs hardly exceeded 15%, but rather to some intrinsic mRNA property (*Hubstenberger et al., 2017*). In this respect, the frequency of low usage codons correlated more with PB localization ($r_s = 0.59$, p<0.0001) than with protein yield ($r_s = -0.21$, p<0.0001, significantly different from 0.59, p<0.0001) (*Figure 2—figure supplement 1E*). Conversely, the CDS length correlated more with protein yield ($r_s = -0.43$, p<0.0001) than with PB localization ($r_s = 0.26$, p<0.0001, significantly different from $-0.43$, p<0.0001). Nevertheless, the length of the CDS and its GC content contributed independently to PB localization (*Figure 2E*). Finally, combining the frequency of low usage codons with the CDS length, that is, considering the absolute number of low usage codons per CDS, was a shared parameter of both protein yield ($r_s = -0.46$, p<0.0001, *Figure 2F*) and PB localization ($r_s = 0.49$, p<0.0001). Strikingly, CDS with more than 100 low usage codons were particularly enriched in PBs, while those under 100 were mostly excluded (*Figure 2F*). One of the mechanisms linking codon usage to translation yield could be the abundance of cognate tRNAs (*Novoa and Ribas de Pouplana, 2012*). However, codon usage in PB-excluded mRNAs was not more adapted to the abundance of amino-acylated tRNAs (*Evans et al., 2017*) than codon usage in PB-enriched mRNAs (*Figure 2—figure supplement 2*). In conclusion, the strong GC bias in PB mRNAs results in both a biased amino acid usage in encoded proteins and a biased codon usage. Furthermore, the high number of low usage codons in PB mRNAs is a likely determinant of their low protein yield.

## The PB assembly factor DDX6 has opposite effects on mRNA stability and translation rate depending on their GC content

In human, the DDX6 RNA helicase is key for PB assembly (*Minshall et al., 2009*). It associates with a variety of proteins involved in mRNA translation repression and decapping (*Ayache et al., 2015*; *Bish et al., 2015*), suggesting that it plays a role in both processes. To investigate how DDX6 activity is affected by mRNA GC content, we conducted a polysome profiling experiment in HEK293 cells transfected with DDX6 or control β-globin siRNAs for 48 hr. In these conditions, DDX6 expression decreased by 90% compared to control cells (*Figure 3—figure supplement 1A*). The polysome profile was largely unaffected by DDX6 silencing, implying that DDX6 depletion did not grossly disturb global translation (*Figure 3—figure supplement 1B*). Polysomal RNA isolated from the sucrose

gradient fractions (*Figure 3—figure supplement 1B*) and total RNA were used to generate libraries using random hexamers to allow for poly(A) tail-independent amplification. As expected, both total and polysomal DDX6 mRNA was markedly decreased (by 72%) following DDX6 silencing (*Figure 3—figure supplement 1C–E*; *Supplementary file 1*, sheet1). Since DDX6 is cytoplasmic (*Ernoult-Lange et al., 2009*) and has a role in mRNA decay, we assumed that changes in total mRNA accumulation generally reflected an increased stability of the transcripts, though we cannot exclude altered transcription levels for some of them. As polysomal accumulation can result from both regulated translation and a change in total RNA without altered translation, we then used the polysomal to total mRNA ratio as a proxy measurement of translation rate. Nevertheless, for few transcripts, polysomal enrichment may reflect an elongation block rather than an increased rate of initiation. Analysis of the whole transcriptome showed a link between mRNA fate following DDX6 depletion and their GC content, but, intriguingly, the correlation was positive for changes in total RNA ($r_s$ = 0.45, p<0.0001; *Figure 3—figure supplement 1F*) and negative for changes in polysomal RNA ($r_s$ = −0.32, p<0.0001; *Figure 3—figure supplement 1G*). Therefore, DDX6 depletion affected different mRNA subsets in total and polysomal RNA.

The extent of mRNA stabilization steadily increased with the GC content and became predominant for transcripts with >50% GC (*Figure 3A*, left panel, *Figure 3—figure supplement 2A*). This analysis was repeated on an independent dataset available from the ENCODE project (*ENCODE Project Consortium, 2012*), obtained in a human erythroid cell line, K562, following induction of a stably transfected DDX6 shRNA, and using an oligo(dT)-primed library. Despite the differences in cell type, depletion procedure and sequencing methods, again, mRNA stabilization preferentially concerned those with high GC content ($r_s$ = 0.59, p<0.0001; *Figure 3A*, right panel, *Figure 3—figure supplement 2A*; *Supplementary file 1*, sheet2). In contrast, following DDX6 silencing in HEK293 cells, the translation rate predominantly increased for transcripts with less than 45% GC ($r_s$ = −0.53, p<0.0001; *Figure 3B*, *Figure 3—figure supplement 2A*). As a result, mRNAs with the most upregulated translation rate were the least stabilized, and conversely (*Figure 3—figure supplement 2B*).

To investigate how DDX6 activity was related to its binding to RNA, we used the CLIP dataset of K562 cells, also available from the ENCODE project. In both HEK293 and K562 cells, the mRNAs clipped to DDX6 were particularly stabilized after DDX6 knockdown, as compared to all mRNAs

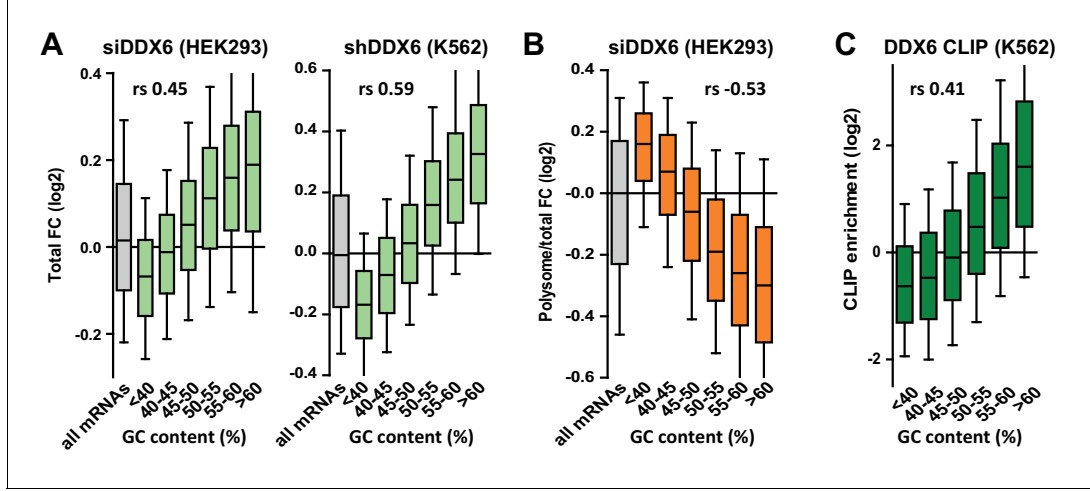

**Figure 3.** DDX6 silencing has opposite effects on mRNA fate depending on their GC content. (**A**) mRNA stabilization after DDX6 silencing in HEK293 and K562 cells applies to GC-rich mRNAs. The fold-changes (FC) in mRNA accumulation were analyzed as in *Figure 1B*. (**B**) mRNA translation derepression after DDX6 silencing in HEK293 cells applies to AU-rich mRNAs. The fold-changes in translation rate (polysomal/total mRNA ratio) were analyzed as in (**A**). (**C**) GC-rich mRNAs are particularly enriched in the DDX6 CLIP experiment.

The online version of this article includes the following figure supplement(s) for figure 3:

**Figure supplement 1.** Polysome profiling following DDX6 silencing.
**Figure supplement 2.** Impact of DDX6 binding and mRNA length on DDX6 dependency.
**Figure supplement 3.** Impact of the GC content on DDX6-dependency.

(*Figure 3—figure supplement 2C*; *Supplementary file 1*, sheet3), while they were not translationally derepressed in HEK293 cells (*Figure 3—figure supplement 2D*). In agreement, mRNAs with a high GC content were preferentially enriched in the DDX6 CLIP experiment ($r_s$ = 0.41, p<0.0001; *Figure 3C*, *Figure 3—figure supplement 2A*). Then, as we previously showed that DDX6 can oligomerize along repressed transcripts (*Ernoult-Lange et al., 2012*), we also considered mRNA length. While DDX6-dependent decay had a marginal preference for short transcripts ($r_s$ = −0.09, p<0.0001), as a combined effect of CDS and 3'UTR length (*Figure 3—figure supplement 2E,F*), DDX6-dependent translation repression was independent of the CDS length but higher on mRNAs with long 3'UTRs ($r_s$ = 0.16, p<0.0001; *Figure 3—figure supplement 2E,G*). Interestingly, the GC content of the CDS and the 3'UTR were similarly predictive of DDX6 sensitivity, whether for mRNA stability ($r_s$ = 0.42 and 0.40 for CDS and 3'UTR, respectively, p<0.0001 for both) or for translation repression ($r_s$ = −0.53 and −0.52, respectively, p<0.0001 for both), while the 5'UTR was less significant ($r_s$ = 0.18 and −0.15 for stability and translation repression, respectively, p<0.0001 for both; *Figure 3—figure supplement 3A–C*).

Altogether, we showed that DDX6 knockdown affected differentially the mRNAs depending on the GC content of both their CDS and 3'UTR, with the most GC-rich mRNAs being preferentially regulated at the level of stability and the most AU-rich mRNAs at the level of translation.

## DDX6/XRN1 and PAT1B decrease the stability of separate sub-classes of mRNAs with distinct GC content

DDX6 acts as an enhancer of decapping to stimulate mRNA decay, upstream of RNA degradation by the XRN1 5'−3' exonuclease. To investigate whether XRN1 targets are similarly GC-rich, we performed XRN1 silencing experiments in two cell lines. HeLa cells were transfected with XRN1 siRNA (*Figure 4—figure supplement 1A*; *Supplementary file 1*, sheet4), while HCT116 cells stably transfected with an inducible XRN1 shRNA were induced with doxycyclin (*Figure 4—figure supplement 1B*; *Supplementary file 1*, sheet5), both for 48 hr. In both cell lines, XRN1-dependent decay preferentially acted on mRNAs which were GC-rich ($r_s$ = 0.41 for HeLa and 0.49 for HCT116, p<0.0001 for both; *Figure 4A*, *Figure 4—figure supplement 1A*) and localized out of PBs ($r_s$ = −0.35, p<0.0001; *Figure 4—figure supplement 1C*), as observed for DDX6.

PAT1B is a well-characterized direct DDX6 partner known for its involvement in mRNA decay (*Vindry et al., 2017*; *Braun et al., 2010*; *Ozgur et al., 2010*; *Vindry et al., 2019*). As for DDX6, we assume that changes in steady-state mRNAs following PAT1B silencing generally reflect their increased stability (though, again, we cannot exclude some changes at the transcription level). However, using our previous PAT1B silencing experiment in HEK293 cells (*Vindry et al., 2017*), we surprisingly found a negative correlation between mRNA stabilization after PAT1B and after DDX6 silencing ($r_s$ = −0.31, p<0.0001; *Figure 4—figure supplement 1D*; *Supplementary file 1*, sheet6), suggesting that they largely target separate sets of mRNAs. Unexpectedly, the correlation was however positive with translational derepression after DDX6 silencing ($r_s$ = 0.45, p<0.0001; *Figure 4—figure supplement 1E*), indicating that PAT1B preferentially targets mRNAs that are translationally repressed by DDX6. Accordingly, these transcripts are prone to PB storage ($r_s$ = 0.49, p<0.0001; *Figure 4—figure supplement 1F*), as reported previously (*Vindry et al., 2017*). Indeed, in contrast to DDX6 and XRN1 decay targets, PAT1B targets tended to be AU-rich ($r_s$ = −0.50, p<0.0001; *Figure 4B*, *Figure 4—figure supplement 1A*). To gain insight into the mechanism of regulation by PAT1B, we analyzed the read coverage in the PAT1B silencing experiment (*Figure 4C*) and found it to be unchanged over the whole transcriptome. In contrast, following XRN1 silencing, the 5' coverage was higher, confirming that such an analysis can reveal 5' decay (*Figure 4D*). Of note, in control cells PAT1B target mRNAs had a higher 5' coverage than average (*Figure 4C*), while XRN1 targets had a lower 5' coverage than average (*Figure 4D*). These results suggest that mRNA accumulation in the absence of PAT1B does not result from their 5' end protection.

In conclusion, DDX6 and PAT1B decrease the stability of distinct mRNA subsets, which strongly differ in their GC content. The results suggest that DDX6 is a cofactor of XRN1 5'−3' exonuclease, whereas PAT1B affects 3' to 5' degradation.

To obtain a global visualization of the results we conducted a clustering analysis of the various datasets (*Figure 4E*). Note that to avoid clustering interdependent datasets, we included the changes in polysomal RNA after DDX6 silencing rather than in the polysomal/total RNA ratio. Altogether, the heatmap shows that GC-rich mRNAs are excluded from PBs and tend to be decayed by

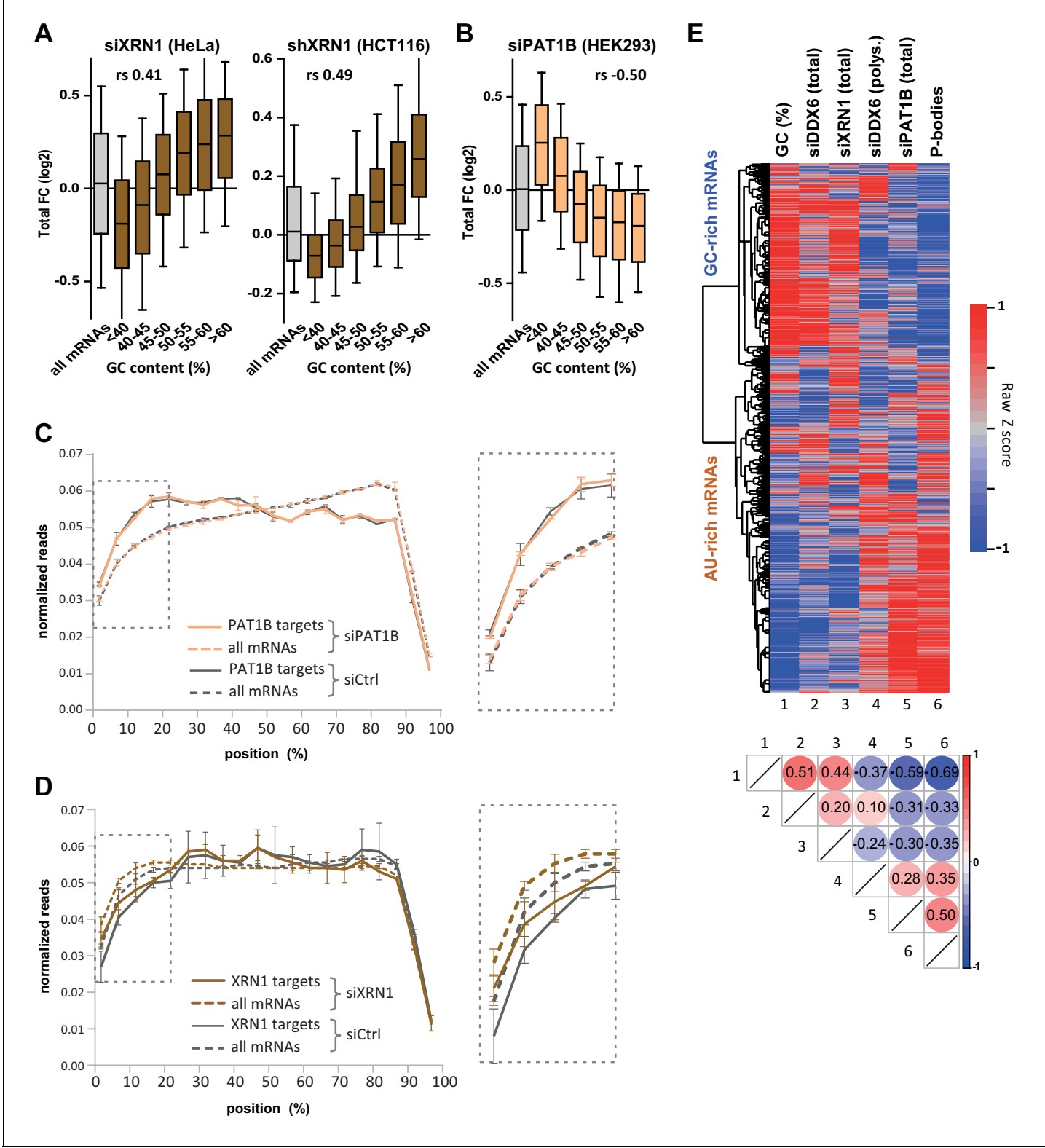

**Figure 4.** XRN1 and PAT1B targets have distinct GC content. (A) mRNA stabilization after XRN1 silencing in HeLa and HCT116 cells applies to GC-rich mRNAs. The analysis was performed as in *Figure 1B*. The GC content distribution for all mRNAs is presented for comparison (in gray). (B) mRNA stabilization after PAT1B silencing in HEK293 cells applies to AU-rich mRNAs. The analysis was performed as in (A). (C) Read coverage of PAT1B targets (FC >0.7, n = 330, solid lines) and all mRNAs (n = 16000, dashed lines) in the siPAT1B dataset. The read coverage was analyzed in each duplicate experiment and normalized as described in Materials and methods. The average value in control cells (gray lines) and after PAT1B silencing (peach

*Figure 4 continued on next page*

*Figure 4 continued*

lines) was plotted, with the bars representing the duplicate values. An expanded view of the dashed box is presented on the right panel. (D) Read coverage of XRN1 targets (FC >0.8, n = 199, solid lines) and all mRNAs (n = 13760, dashed lines) in the siXRN1 dataset. The data were analyzed as in (C). (E) Clustering analysis of mRNAs depending on their GC content, their differential expression after silencing DDX6, XRN1 or PAT1B, and their enrichment in PBs. Raw GC content and log2 transformed ratio of the other datasets were used for the clustering of both transcripts (lines) and datasets (columns). The values were color-coded as indicated on the right scale, and the Spearman correlation matrix is presented below (all p<10$^{-48}$). The heatmap highlights the distinct fate of GC-rich and AU-rich mRNAs.

The online version of this article includes the following figure supplement(s) for figure 4:

**Figure supplement 1.** Transcriptome analysis following XRN1 and PAT1B silencing.

a mechanism involving DDX6 and XRN1, while AU-rich mRNAs are recruited in PBs, they undergo DDX6-dependent translation repression and their stability depends on PAT1B.

## Specific mRNA decay factors and translation regulators target mRNAs with distinct GC content

Having shown that GC content is a distinctive feature of DDX6 and XRN1 versus PAT1B targets, we investigated the link between this global sequence determinant and a variety of sequence-specific post-transcriptional regulators for which relevant genome-wide datasets are available (*Figure 5—figure supplement 1A*).

On the mRNA decay side (group I lists), we considered the Nonsense Mediated Decay (NMD) pathway, taking as targets the mRNAs cleaved by SMG6 (*Schmidt et al., 2015*), and the m$^6$A-associated decay pathways, using the targets of the YTHDF2 reader defined by CLIP (*Wang et al., 2014*; *Yang et al., 2015*). We also analyzed mRNAs with a 5'UTR-located G4 motif, which have been shown to be preferential substrates of murine XRN1 in vitro (*Bashkirov et al., 1997*). On the translation regulation side (group II lists), we analyzed the TOP mRNAs, whose translation is controlled by a TOP motif at the 5' extremity (*Thoreen et al., 2012*), and targets of various PB proteins and/or DDX6 partners (*Hubstenberger et al., 2017*; *Ayache et al., 2015*): FXR1-2, FMR1, PUM1-2, IGF2BP1-3, the helicase MOV10, ATXN2, 4E-T, ARE-containing mRNAs and the targets of the two ARE-binding proteins HuR and TTP. We also included mRNAs with a CPE motif, since DDX6 is a component of the CPEB complex that binds CPEs (*Minshall et al., 2007*). Of note, among the group II factors, some are known to also affect mRNA half-life, as exemplified by the ARE-binding proteins (*Wells et al., 2017*). G4, ARE and CPE motifs have been defined in silico, while the targets of the various factors originate from RIP and CLIP approaches in human cells or mouse studies in the case of TOP mRNAs (see Materials and methods).

Intriguingly, compared to all mRNAs, group I list mRNAs were GC-rich, as well as TOP mRNAs and ATXN2 targets, whereas all other group II lists were AU-rich (*Figure 5A*). Furthermore, they shared common behavior in the various experiments. This is summarized in *Figure 5B* in a heatmap representing their median value in each dataset, while *Figure 5—figure supplements 1* and *2* provide detailed analysis, as described below.

Group I list mRNAs tended to be dependent on DDX6 and XRN1 but not on PAT1B for stability (*Figure 5—figure supplement 1B–D*), with nevertheless some variation between cell lines, as only SMG6 targets were sensitive to XRN1 depletion in HeLa cells (*Figure 5—figure supplement 1C*, upper panel). They did not accumulate in PBs and their translation rate was independent of DDX6 (*Figure 5—figure supplement 2A,B*). These results were consistent with their high GC content and our global analysis above. However, surprisingly, within PB-excluded mRNAs, there was little or no additional effect of being a SMG6 target, an YTHDF2 target or containing a G4 motif, neither for DDX6- nor for XRN1-dependent decay (*Figure 5—figure supplement 2C,D*).

Group II list mRNAs, except TOP mRNAs, ATXN2 and 4E-T targets, had the exact mirror fate compared to group I lists: they were stabilized following PAT1B silencing (*Figure 5—figure supplement 1D*), as previously reported for ARE-containing mRNAs and the targets of the ARE-BPs HuR and TTP (*Vindry et al., 2017*), but not following DDX6 or XRN1 silencing (*Figure 5—figure supplement 1B,C*); they were enriched in PBs and translationally more active after DDX6 silencing (*Figure 5—figure supplement 2A,B*), which is consistent with the reported presence of most of these regulatory proteins in PBs (*Hubstenberger et al., 2017*; *Franks and Lykke-Andersen, 2007*).

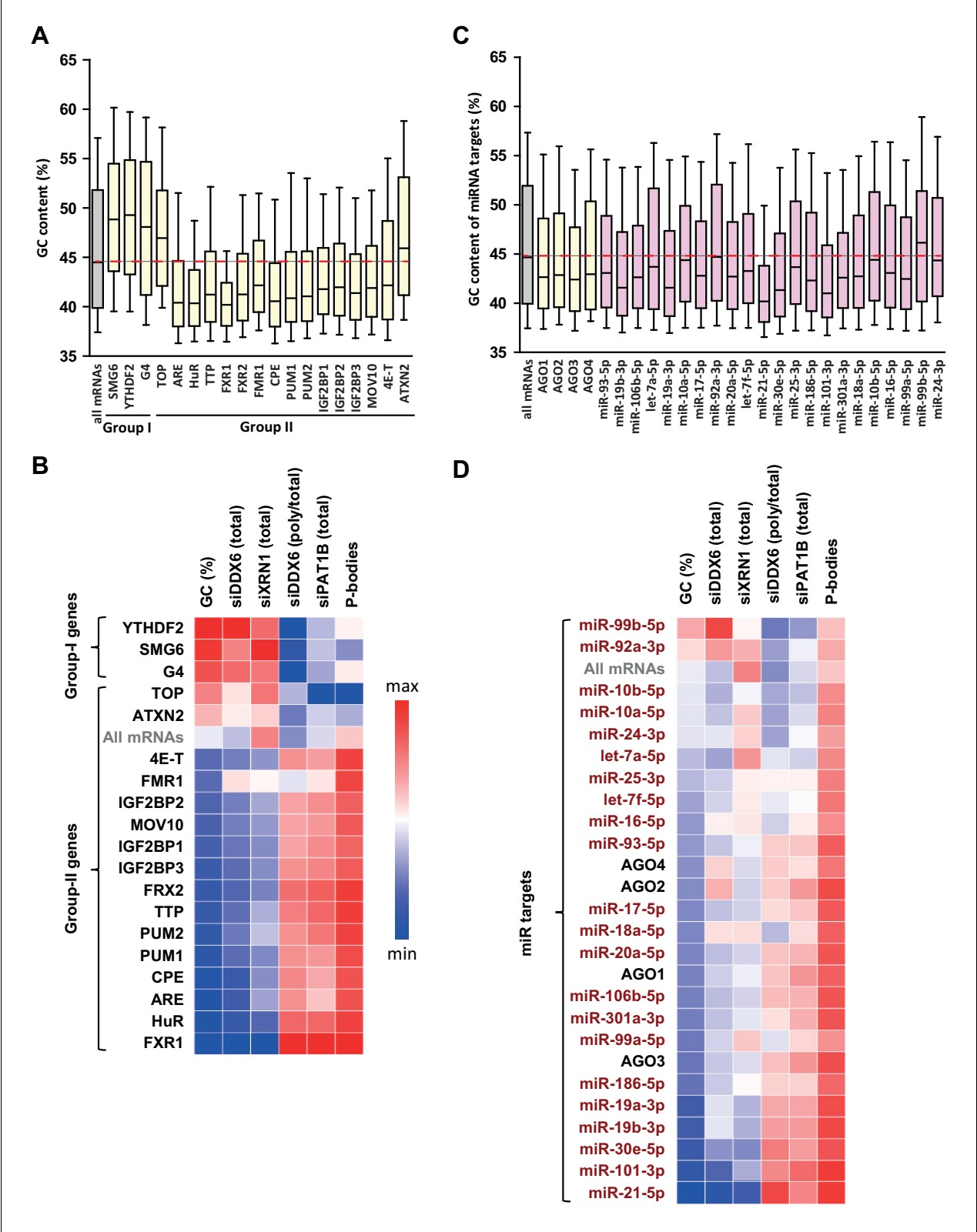

**Figure 5.** GC biases in the targets of various RNA decay factors, translation regulators and miRNAs. (**A**) GC content biases in the targets of various RBPs. The targets of the indicated factors were defined using CLIP experiments or motif analysis (see Materials and methods). The boxplots represent the distribution of the GC content of their gene. The distribution for all mRNAs is presented for comparison (in gray) and the red dashed line indicates its median value. (**B**) Heatmap representation of the different factors depending on the behavior of their mRNA targets in the different datasets. The
*Figure 5 continued on next page*

*Figure 5 continued*

lines were ordered by increasing GC content, and the columns as in *Figure 4E*. (C) GC content biases in the targets of various AGO proteins and miRNAs. The AGO targets (in yellow) were defined using CLIP experiments and the miRNAs targets (in violet) using miRTarbase. The data are represented as in (A). (D) Heatmap representation of AGO and miRNAs depending on the behavior of their mRNA targets in the different datasets. The data were represented as in (B), using the same color code.

The online version of this article includes the following figure supplement(s) for figure 5:

**Figure supplement 1.** Targets of group I and II regulators (part I).
**Figure supplement 2.** Targets of group I and II regulators (part II).
**Figure supplement 3.** Targets of the miRNA pathway.
**Figure supplement 4.** Targets of the group I and II regulators behave like mRNAs of similar GC content.
**Figure supplement 5.** miRNA targets behave like mRNAs of similar GC content.

Among the three group II outsiders, ATXN2 targets and TOP mRNAs behaved like group I lists, except that they were not dependent on DDX6 for stability. ATXN2 is a major DDX6 partner, but one which is excluded from PBs (*Ayache et al., 2015*; *Nonhoff et al., 2007*), consistent with its targets also being excluded from PBs (*Figure 5—figure supplement 2A*). However, these mRNAs were only weakly or not stabilized following DDX6 or XRN1 silencing (*Figure 5—figure supplement 1B,C*), leaving unresolved the function of the ATXN2/DDX6 interaction in the cytosol. TOP mRNAs are special within group II lists, in that their translational control relies on a cap-adjacent motif rather than on 3'UTR binding. 4E-T is a major DDX6 partner required for PB assembly (*Ayache et al., 2015*; *Kamenska et al., 2016*). Its targets were markedly enriched in PBs (*Figure 5—figure supplement 2A*), though, intriguingly, poorly affected by PAT1B silencing in terms of stability (*Figure 5—figure supplement 1D*), or by DDX6 silencing in terms of translation (*Figure 5—figure supplement 2B*). This dissociation between PB localization and mRNA fate indicated that PB recruitment is not sufficient for an mRNA to have a PAT1B-dependent stability and DDX6-dependent translation repression. It also pointed to a particular role of 4E-T in PB targeting or scaffolding.

Our analysis is also informative on the link between DDX6-dependent decay and codon usage. Previous yeast studies have debated whether suboptimal codons could enhance DDX6 recruitment to trigger mRNA decay (*Radhakrishnan et al., 2016*) or not (*Chan et al., 2018*; *He et al., 2018*). We showed above that GC-rich mRNAs, which tend to be decayed by DDX6 and to bind DDX6 (*Figure 3A,C*), are enriched for optimal rather than suboptimal codons (*Figure 2D*). Furthermore, the correlation between polysomal retention (defined as the fraction of total mRNA present in polysomes) and DDX6-dependent decay was weak ($r_s$ = 0.10, p<0.0001; *Figure 5—figure supplement 2E*). Thus, in HEK293 cells, this mechanism seemed to account for a minor part of DDX6-dependent decay, if any, as also found in mouse stem cells (*Freimer et al., 2018*).

In conclusion, we observed that mRNA decay regulators preferentially target GC-rich mRNAs, which undergo DDX6- and XRN1-dependant decay, whereas most translation regulators preferentially target AU-rich mRNAs, which are subjected to storage in PBs and have a PAT1B-dependent stability .

## Targets of the miRNA pathway have a biased GC content

The miRNA pathway, which leads to translation repression and mRNA decay, has been previously associated with DDX6 activity and PB localization (*Bhattacharyya et al., 2006*; *Chu and Rana, 2006*). To study this pathway, we used the list of AGO1-4 targets, as identified in CLIP experiments (*Yang et al., 2015*) (*Figure 5—figure supplement 3A*). In addition, we analyzed the experimentally documented targets of the 22 most abundant miRNAs in HEK293 cells (19 from *Hafner et al., 2010*, and three additional ones from our own quantitation, *Figure 5—figure supplement 3B*), as described in miRTarBase (*Hsu et al., 2014*). The mRNA targets of AGO proteins were AU-rich, as observed for most group II RBPs, and this was also true for the targets of most miRNAs when analyzed separately (*Figure 5C*). Overall, they also shared common behavior in the various silencing experiments and PB dataset, with nevertheless some differences. This is summarized in *Figure 5D* in a heatmap representing their median value in each dataset, while *Figure 5—figure supplement 3* provides detailed analysis, as described below.

The AGO targets tended to accumulate in PBs (*Figure 5—figure supplement 3C*; note that the number of AGO4 targets was too small to reach statistical significance) and their translation rate was

DDX6-dependent (*Figure 5—figure supplement 3D*). In terms of stability, only AGO2 targets were marginally DDX6-dependent (*Figure 5—figure supplement 3E*), and the effects were not stronger when analyzing separately the mRNAs enriched or excluded from PBs (*Figure 5—figure supplement 3F*). In contrast, their stability was PAT1B-dependent (*Figure 5—figure supplement 3G*).

The targets of the 22 miRNAs had a behavior overall similar to the targets of AGO proteins, with accumulation in PBs, DDX6-dependent translation, and PAT1B rather than DDX6- or XRN1-dependent stability (*Figure 5D*). However, our analysis revealed some differences between miRNAs, particularly in terms of extent of PB storage (*Figure 5—figure supplement 3H*), which appeared associated with distinct GC content: at the two extremes, miR21-5p targets were particularly AU-rich and strongly enriched in PBs, while the targets of miR-99b-5p, the most GC-rich in these sets, were not. In terms of translation, miR-18a-5p targets were not sensitive to DDX6 silencing, despite clear enrichment in PBs. In terms of stability, the targets of the less GC-rich miR-99b-5p were sensitive to DDX6 but not PAT1B silencing.

In conclusion, miRNA targets generally tend to be AU-rich, like the targets of most translation regulators, and accumulate in PBs. While their translation depends on DDX6, their stability is not markedly affected following DDX6 or XRN1 silencing, but is dependent on PAT1B.

## The GC content of mRNAs shapes post-transcriptional regulation

As the global GC content appeared closely linked to mRNA fate, but also to RBP and miRNA binding, as well as to translation activity, our analyses then aimed at ranking the importance of these various features.

We first assessed the respective weight of the GC content and the binding capacity of particular RBPs. To this aim, we binned the whole transcriptome depending on its GC content (bin size of 500 transcripts). The median fold-changes of the bins in each RNAseq dataset were calculated and plotted as a function of their median GC content. Median values were similarly calculated for the various group I and II target lists and overlaid for comparison (*Figure 5—figure supplement 4*). Surprisingly, the fold changes of the targets of particular RBPs generally fell very close to the tendency plot based on GC content only. This was particularly true for DDX6- and XRN1-dependent decay, with only 4E-T targets being more stabilized after DDX6 silencing than expected from their GC content (*Figure 5—figure supplement 4A,B*). In terms of translation rate, only FXR1 targets were slightly more translated after DDX6 depletion than expected from their GC content (*Figure 5—figure supplement 4C*). FXR1 targets were also more dependent on PAT1B for stability (*Figure 5—figure supplement 4D*) and more enriched in PBs (*Figure 5—figure supplement 4E*). In the case of HuR, TTP, FXR1-2, FMR1, PUM2, IGF2BP1-3 and MOV10, there was some, but minimal, additional PAT1B-sensitivity and PB enrichment.

Similarly, the fate of the miRNA targets was mostly in the range expected from their GC content (*Figure 5—figure supplement 5*). Nevertheless, some miRNA-specific effects were observed. For instance, the targets of several miRNAs were more stabilized than expected after DDX6 depletion, including miR-99b-5p, 92a-3p, 16–5 p, 18a-5p, 19a-3p, 19b-3p (*Figure 5—figure supplement 5A*), though this was not observed following XRN1 depletion (*Figure 5—figure supplement 5B*). Similarly, while the targets of miR-101–3 p and miR-21–5 p were both particularly enriched in PBs (*Figure 5—figure supplement 5E*), only miR-101–3 p targets were particularly dependent on PAT1B for stability (*Figure 5—figure supplement 5D*). Interestingly, we noted that the median GC content of the miRNA targets correlated with the GC content of the miRNA itself (*Figure 5—figure supplement 5F*). Thus, despite their small size, the miRNA binding sites tend to have a GC content similar to that of their full-length host mRNA, which affects their fate in terms of PB localization and post-transcriptional control. Altogether, our analysis showed that, in steady-state conditions, mRNA GC content is a major parameter in terms of PB localization and regulation by DDX6, XRN1 and PAT1B, while the presence of binding sites for regulatory proteins makes subsidiary contributions.

Next, the strong correlation observed between GC content and PB localization raised the possibility that localization out of PBs was sufficient to determine mRNA sensitivity to DDX6 and XRN1 decay. While a tempting hypothesis, it could not explain all DDX6- and XRN1-dependent decay, since TOP mRNAs were strongly excluded from PBs (*Figure 5—figure supplement 2A*), but unaffected by DDX6 or XRN1 silencing (*Figure 5—figure supplement 1B,C*). To address this issue more generally, we considered the fate of the minor subset of AU-rich mRNAs that were excluded from PBs. Compared to other similarly AU-rich transcripts, these mRNAs were indeed more sensitive to

XRN1-dependent decay (*Figure 6—figure supplement 1A*). However they were not more sensitive to DDX6-dependent decay (*Figure 6—figure supplement 1B*). This suggested that XRN1 preference for GC-rich mRNAs is at least in part related to their exclusion from PBs, whereas DDX6 has a true preference for GC-rich mRNAs.

Interestingly, these PB-excluded AU-rich mRNAs were strongly enriched in mRNAs encoding secreted proteins and proteins associated with membranous organelles, with GO categories related to mitochondria, intracellular organelles and extracellular matrix proteins representing up to 36% of the transcripts (*Figure 6—figure supplement 1C*). Thus, while mRNA localization in PBs is highly influenced by their GC content, it may also be outcompeted by retention on membranous organelles and plasma membrane.

## Contribution of both the CDS and 3'UTR GC content to PB localization

The next major issue was to distinguish which of the CDS or 3'UTR is more important for PB localization, since they have very similar GC contents ($r_s$ = 0.72, p<0.0001).

As a first approach, we analyzed PB localization of long non-coding RNAs (lncRNAs) (*Hubstenberger et al., 2017*). The correlation between their GC content and PB accumulation was significant ($r_s$ = −0.20, p<0.0001), but much weaker than that observed for mRNAs (−0.64, *Figure 1B*) or 3'UTRs (−0.55, *Figure 1D*) (−0.20 and −0.55 are significantly different, p<0.0001). In fact, AU-rich lncRNAs poorly accumulated in PBs, while GC-rich lncRNAs were excluded (*Figure 6—figure supplement 1D,E*). This suggested that the coding capacity of mRNAs was important for PB localization. As a second approach, we directly analyzed the respective contribution of the GC content of CDS and 3'UTR to PB localization. On one side, we analyzed transcripts by groups of similar 3'UTR GC content. Their GC3 was systematically much lower in PB mRNAs than in PB-excluded mRNAs, with differences ranging between 9% and 13% GC (*Figure 6A*, *Figure 6—figure supplement 1E*). In a mirror analysis, we analyzed groups of transcripts with similar GC3. The importance of the 3'UTR GC content became visible only for GC3 higher than 50% GC (note that GC3 median value is 59% GC), with AU-rich 3'UTR allowing for their accumulation in PBs despite a GC-rich CDS (*Figure 6B*, *Figure 6—figure supplement 1E*). We concluded that both the CDS and the 3'UTR GC content are important for PB localization, with the CDS being the primary feature.

We speculate that suboptimal translation of AU-rich CDS makes mRNAs optimal targets for translation regulation, since any control mechanism has to rely on a limiting step. Conversely, optimally translated transcripts would be better controlled at the level of stability. One prediction is that proteins produced in limiting amounts, such as those encoded by haplo-insufficiency genes, are more likely to be encoded by PB mRNAs. Genome-wide haplo-insufficiency prediction scores have been defined for human genes, using diverse genomic, evolutionary, and functional properties trained on known haplo-insufficient and haplo-sufficient genes (*Huang et al., 2010*; *Steinberg et al., 2015*). Using these scores, we found that haplo-insufficient mRNAs were indeed significantly enriched in PBs (*Figure 6C*).

To add experimental support to the importance of GC content for PB assembly, we conducted two assays. First, we analyzed the localization of reporter transcripts that differ only by the GC content of their CDS. HEK293 cells stably expressing the PB marker GFP-LSM14A (*Hubstenberger et al., 2017*) were transfected with plasmids containing an AU-rich (36% GC) or GC-rich (58% GC) CDS that encodes the same Renilla luciferase (Rluc) protein. After 24 h cells were analyzed for luciferase activity and transcript localization. In agreement with our previous analyses, Rluc protein yield was considerably reduced (4.5-fold) using the AU-rich rather than the GC-rich version of the CDS, despite similar mRNA levels (*Figure 6D*). The localization of the Rluc transcripts was then analyzed by smiFISH using AU-rich or GC-rich specific probes (*Figure 6—figure supplement 2A*, *Supplementary file 2*) (*Tsanov et al., 2016*). PBs containing clusters of Rluc mRNA molecules were five times more frequent using the AU-rich than the GC-rich version of the CDS (*Figure 6E,F*). A similar result was obtained in HEK293 cells after PB immunostaining with DDX6 antibodies (*Figure 6F*, *Figure 6—figure supplement 2B*). Therefore, simply changing the GC content of this medium-size CDS (564 codons) was sufficient to modify mRNA localization in PBs.

Second, we tested the capacity of AU-rich and GC-rich RNA to form granules independently of translation. To this aim, we set-up a cell-free assay using HEK293 cells expressing GFP-LSM14A to monitor the formation of fluorescent PB-like granules and count them by flow cytometry, as previously performed for PBs (*Hubstenberger et al., 2017*). After lysis and elimination of preexisting PBs

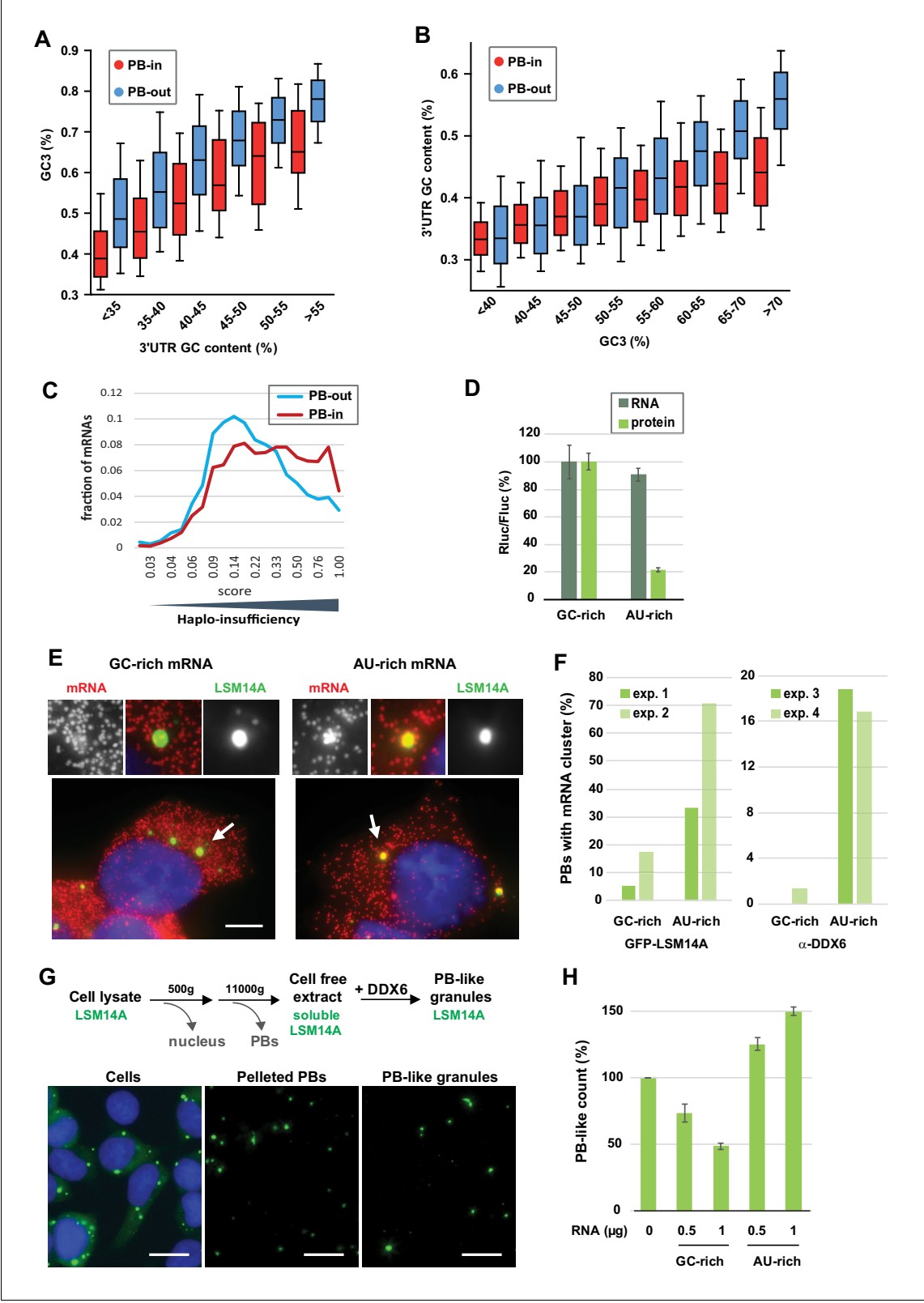

**Figure 6.** The GC content of the CDS and the 3'UTR both contribute to PB localization. (**A**) General importance of the CDS. Transcripts were subdivided into six classes depending on the GC content of their 3'UTR (from <40 to >55%). The boxplots represent the distribution of their CDS GC content at position 3 (GC3) in PB-enriched (PB-in) and PB-excluded (PB-out) mRNAs. (**B**) Importance of the 3'UTR for GC-rich CDSs. Transcripts were subdivided into eight classes depending on their GC3 (from <40 to >70%). The boxplots represent the distribution of their 3'UTR GC content in PB-

*Figure 6 continued on next page*

*Figure 6 continued*

enriched (PB-in) and PB-excluded (PB-out) mRNAs. (C) The transcripts of haplo-insufficiency genes are enriched in PBs. The haplo-insufficiency score is the probability that a gene is haplo-insufficient, as taken from the *Huang et al. (2010)* study. The analysis was performed for PB-enriched (PB-in, n = 4646, median score 0.26) and PB-excluded (PB-out, n = 4205, median score 0.17) mRNAs. The difference of distribution of haplo-insufficiency scores was statistically significant using a two tail Mann-Whitney test: p<0.0001. The results were similar using *Steinberg et al. (2015)* scores. (D) Protein yield is higher from a GC-rich than an AU-rich CDS. HEK293 cells were transfected with Rluc reporters differing by the GC content of their CDS, along with a control Fluc plasmid. After 24 hr, mRNA levels were measured by qPCR and protein levels by luciferase activity. The Rluc to Fluc ratio for the GC-rich reporter was set to 100 (n = 3). Error bars, SD. (E, F) Preferential localization of AU-rich transcripts in PBs. HEK293 cells expressing GFP-LSM14A were transfected with the AU-rich and GC-rich Rluc reporters and the localization of the Rluc transcripts (in red) was analyzed by smiFISH. Representative cells are shown in (E). Bar, 5 μm. Arrows indicate the PBs enlarged above. The experiment was performed in duplicate (exp. 1 and 2) and repeated in HEK293 cells where PBs were immunostained using DDX6 antibodies (exp. 3 and 4). The percentage of PBs containing clusters of Rluc transcripts in the four experiments is represented in (F). Exp.1: 56/75 PBs from 21/27 cells; exp.2: 87/75 PBs from 38/35 cells; exp.3: 31/32 PBs from 15/19 cells; exp.4: 72/83 PBs from 34/41 cells (G) Assembly of PB-like granules in cell-free extracts from HEK293 cells expressing GFP-LSM14A. The scheme recapitulates the main steps of the assay. Fluorescence microscopy images show that PBs in cells, PBs after cell lysis, and reconstituted PB-like granules have similar size. Bar, 10 μm. (H) AU-rich RNA favors the formation of PB-like granules. PB-like granules were assembled in cell-free extracts in the presence of AU-rich or GC-rich RNA, and counted by flow cytometry. Their number in the absence of added RNA was set to 100 (n = 3 experiments in duplicate, using two independent cell-free extracts and RNA preparations). Error bars, SD.

The online version of this article includes the following figure supplement(s) for figure 6:

**Figure supplement 1.** Role of PB localization in XRN1 and DDX6 sensitivity and importance of the coding property for PB localization.
**Figure supplement 2.** The GC content of reporters RNAs is key for PB localization.

by centrifugation, addition of recombinant DDX6 triggered the formation of new granules on ice, in a dose-dependent manner (*Figure 6—figure supplement 2C–E*). These granules had a similar size to endogenous PBs (*Figure 6G*, *Figure 6—figure supplement 2D*). This reconstitution assay was surprisingly efficient, as granule formation required rather low concentrations of both the lysate components (about 100-fold lower than in cells, see Materials and methods) and recombinant DDX6 (0.17 μM versus 3.3 μM in cells, *Ernoult-Lange et al., 2012*). Next, the cell-free extract was briefly treated with micrococcal nuclease to decrease the amount of cellular RNA, and the assay was repeated with or without addition of an either AU-rich or GC-rich 1700 nt-long synthetic RNA (*Figure 6H*, *Figure 6—figure supplement 2F*). The AU-rich RNA increased the number of PB-like granules in a dose-dependent manner, while GC-rich RNA prevented their formation. Therefore, in the complex lysate environment and at 0°C, uncapped non-polyadenylated AU-rich RNA specifically favor the condensation of granules that are DDX6-dependent and contain LSM14A, two proteins that play a major role in the assembly of cellular PBs.

We conclude from these experimental data and our previous analyses that both the CDS and the 3'UTR contribute to PB localization. Low GC content in the CDS likely acts, at least in part, through codon usage and low translation efficiency. In the 3'UTR low GC content could allow for the binding of RBPs with affinity for AU-rich motifs and/or influence RNA secondary structure.

## Discussion

### An integrated model of post-transcriptional regulation

Our combined analysis of the transcriptome of purified PBs together with transcriptomes following the silencing of broadly-acting storage and decay factors, including DDX6, XRN1 and PAT1B, provided a general landscape of post-transcriptional regulation in human cells, where mRNA GC content plays a central role. As schematized in *Figure 7*, GC-rich mRNAs are excluded from PBs and mostly controlled at the mRNA level by a mechanism involving the helicase DDX6 and the 5'−3' exonuclease XRN1. In contrast, AU-rich mRNAs are enriched in PBs and rather controlled at the level of translation by a mechanism also involving DDX6, while their accumulation tend to depend on a mechanism involving the DDX6 partner PAT1B and most likely 3' decay. Accordingly, NMD and m6A-associated mRNA decay pathways tend to target GC-rich mRNAs, while most sequence-specific translation regulators and miRNAs tend to target AU-rich mRNAs. The distinct fate of GC-rich and AU-rich mRNAs correlates with a contrasting protein yield resulting from both different codon usage and CDS length. Thus, 5' mRNA decay appears to control preferentially mRNAs with optimal

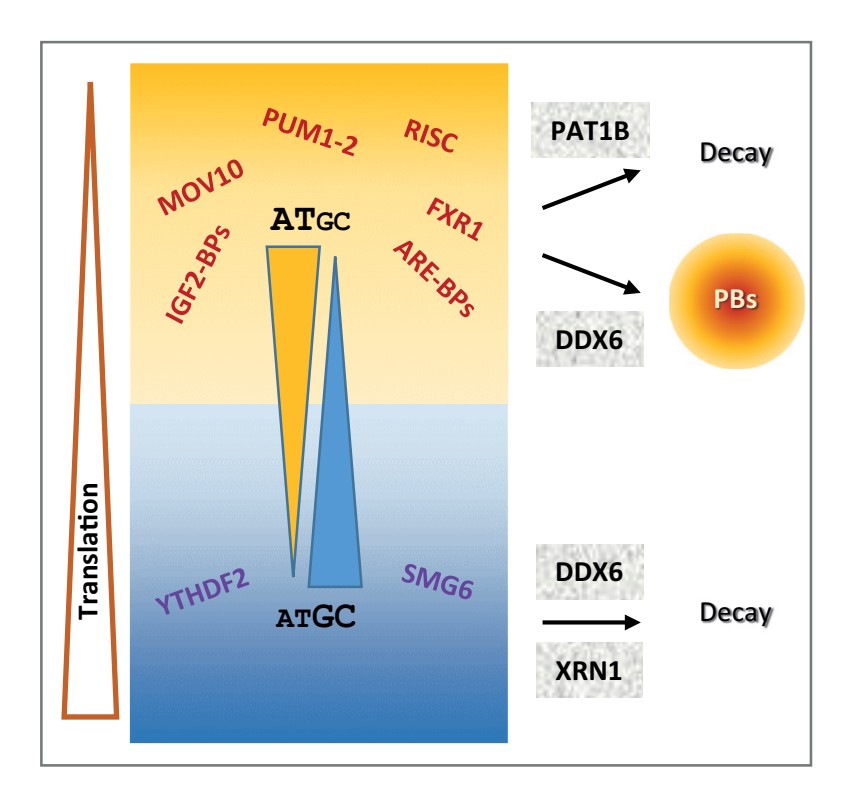

**Figure 7.** Schematic representation recapitulating the features of mRNA post-transcriptional regulation depending on their GC content.
The online version of this article includes the following figure supplement(s) for figure 7:

**Figure supplement 1.** Distribution of the gene GC content in various eukaryotic genomes.

translation, which are mostly GC-rich, whereas translation regulation is mostly used to control mRNAs with limiting translational efficiency, which are AU-rich.

It should be stressed that this model only applies to post-transcriptional regulation pathways that involve PBs, XRN1, DDX6 and PAT1B. Moreover, while the analysis was consistent in proliferating cells of various origins, giving rise to a general model, it is possible that changes in cell physiology, for instance at particular developmental stages or during differentiation, rely on a different mechanism. In addition, our analysis focused on trends common to most transcripts, which does not preclude that particular mRNAs could be exceptions to the general model, being GC-rich and translationally controlled, or AU-rich and regulated by 5' decay. In terms of translation yield and PB localization, this model is strongly supported by our experiments using AU-rich and GC-rich RNAs: AU-rich reporter mRNAs have a low protein yield compared to GC-rich ones, they preferentially localize in PBs in cells, and they enhance the formation of PB-like granules in a cell free extract.

## GC content and codon usage

While the redundancy of the genetic code should enable amino acids to be encoded by synonymous codons of different base composition, the wide GC content variation between PB-enriched and PB-excluded mRNAs has consequences on the amino acid composition of encoded proteins. It also strongly impacts the identity of the wobble base: in PB mRNAs, the increased frequency of A/U at position 3 of the codon mechanically results in an increased use of low usage codons. As CDS are also longer in PB mRNAs, it further increases the number of low usage codons per CDS in these mRNAs. Interestingly, we showed that the absolute number of low usage codons per CDS best correlates with low protein yield. Thus, these results provide a molecular mechanism to a previously unexplained feature of PB mRNAs, that is, their particularly low protein yield, which we reported was

an intrinsic property of these mRNAs and not simply the result of their sequestration in PBs (*Hubstenberger et al., 2017*). Interestingly, the mRNAs of haplo-insufficiency genes, which by definition are expected to have a limited protein yield, are indeed enriched in PBs (*Figure 6C*).

In addition to the GC-dependent codon bias, we also observed some GC-independent codon bias in PB-enriched mRNAs. Interestingly, some important post-transcriptional regulation programs involve codon usage. This was shown for proliferation- versus differentiation-specific transcripts in human cells (*Gingold et al., 2014*) and for maternal mRNA clearance during the maternal-to-zygotic transition in zebrafish, *Xenopus*, mouse, and *Drosophila* (*Bazzini et al., 2016*). Codon usage could also enable the regulation of small subsets of mRNAs, depending on cellular requirements. In man, half of GO sets show more variability in codon usage than expected by chance (*Rudolph et al., 2016*). Based on GO analysis, we previously demonstrated that PB mRNAs tend to encode regulatory proteins, while PB-excluded mRNAs encode basic functions. Furthermore, proteins of the same complexes tended to be encoded by mRNAs co-regulated in or out of PBs, in so-called post-transcriptional regulons (*Hubstenberger et al., 2017*; *Standart and Weil, 2018*). We speculate that specific codon usage could also underlie these post-transcriptional regulons.

## Distinct mechanisms of mRNA stability control

Our analysis distinguished separate modes of stability control, depending on mRNA GC content. Interestingly, our previous analysis of the read coverage of PB and non-PB RNAs alsosuggested the existence of distinct decay pathways related to 3' and 5' extremities, respectively (*Hubstenberger et al., 2017*). Part of the triage towards the PAT1B versus the DDX6/XRN1-dependent pathway could be somehow associated with the capacity of the transcripts to condense in PBs, since, among the AU-rich mRNAs, only the ones enriched in PBs were affected by PAT1B silencing (not shown). Nevertheless, PBs do not directly mediate the triage, since (i) TOP mRNAs were strongly excluded from PBs, but unaffected by either DDX6 or XRN1 silencing, and (ii) while PBs disappeared after DDX6 silencing (*Minshall et al., 2009*), causing the release of AU-rich mRNAs into the cytosol, only GC-rich mRNAs were stabilized.

Focusing on DDX6-dependent decay, the minor impact of the 5'UTR GC content, compared to CDS and 3'UTR, indicates that this helicase is not simply involved in allowing XRN1 access to the 5' end. It is tempting to propose rather that, by unwinding GC-rich double-stranded regions over the entire length of the mRNA, DDX6 facilitates XRN1 progression. UPF1, another RNA helicase involved in mRNA decay, has also been shown to preferentially affect the decay of GC-rich mRNAs (*Imamachi et al., 2017*). The same observation was made for targets of the NMD pathway, which involves UPF1, SMG6 and SMG7 (*Colombo et al., 2017*). Although the bias in these cases was restricted to the 3'UTR regions, it suggests that DDX6 could act in concert with other helicases for decay of GC-rich mRNAs. For AU-rich mRNAs, either such an active unfolding would be dispensable, or it would rely on other helicases that remain to be identified, with potential candidates being those enriched in purified PBs (*Hubstenberger et al., 2017*). Noteworthily, in agreement with the present study in cell lines, skin fibroblasts from patients with DDX6 missense mutations also showed significant accumulation of GC-rich mRNAs (*Balak et al., 2019*).

Concerning XRN1, we showed that it preferentially acts on PB-excluded mRNAs, that is, on GC-rich mRNAs but also, to a lesser extent, on a subset of AU-rich mRNAs (*Figure 6—figure supplement 1A*). While XRN1 is described as a general decay factor, there is evidence that 5' decay shows some specificity in vivo. In *Drosophila*, mutations in XRN1 have specific phenotypes, including wound healing, epithelial closure and stem cell renewal in testes, suggesting that it specifically degrades a subset of mRNAs (*Pashler et al., 2016*). Of particular relevance, a recent study showed that yeast XRN1 associates with ribosomes and decays mRNAs during translation (*Tesina et al., 2019*). If the mechanism is conserved in human, it would explain why XRN1 preferentially acts on GC-rich mRNAs, since they are the most actively translated mRNAs.

Turning to PAT1B, we showed that its silencing did not affect the same mRNAs as DDX6 silencing. A recent yeast study also showed that Dhh1/DDX6 and Pat1/PAT1B decay targets poorly overlap, suggesting the existence of two separate pathways in yeast as well (*He et al., 2018*). However, the underlying mechanisms may differ in the two organisms, since both Pat1 and Dhh1 targets were poorly translated in yeast, while only PAT1B targets were poorly translated in our study. Previous studies reported that tethered PAT1B decreases the abundance of a reporter mRNA in human cells, as a result of enhanced deadenylation and decapping (*Ozgur et al., 2010*; *Kamenska et al., 2014*;

*Totaro et al., 2011*). However, we did not find any significant evidence of PAT1B-enhanced 5' to 3' decay in our read coverage analysis. Interestingly, genome-wide evidence in yeast too suggests that following decapping a significant fraction of the transcripts up-regulated in cells lacking Pat1 or Lsm1 is efficiently decayed from 3' to 5', rather than by the 5'−3' Xrn1 exonuclease (*He et al., 2018*). Moreover, CLIP experiments in yeast showed a preference for Pat1 and Lsm1 binding to the 3' end of mRNAs (*Mitchell et al., 2013*). Altogether, we therefore favor the possibility that the mechanism by which PAT1B affects mRNA stability in human cells relies prominently on 3' to 5' decay. It could involve the CCR4/CNOT deadenylase and the LMS1-7 complexes, as, despite their low abundance or small size, CNOT1 and LSM2/4 had high scores in our previous PAT1B interactome analysis (*Vindry et al., 2017*; *Vindry et al., 2019*).

PAT1B showed a strong preference for AU-rich targets, including those containing AREs. Many studies have demonstrated a link between AREs and mRNA stability, and its striking importance for processes such as inflammation (*Wells et al., 2017*). Most ARE-BPs promote mRNA destabilization while some ARE-BPs, such as HuR (*Lebedeva et al., 2011*; *Mukherjee et al., 2011*) and AUF1 for a subset of mRNAs (*Yoon et al., 2014*), can stabilize mRNAs. Altogether, these observations raise the possibility that ARE-BPs behave either as enhancers or inhibitors of PAT1B activity in mRNA decay. Similarly, the miRNA pathway could activate this PAT1B activity.

## Translation repression and PB accumulation

DDX6 activity in translation repression has been documented in a variety of biological contexts. In *Xenopus* oocytes, DDX6 contributes to the repression of maternal mRNAs, as a component of the well characterized CPEB complex (*Minshall et al., 2007*). In *Drosophila*, Me31B/DDX6 represses the translation of thousands of mRNAs during the early stages of the maternal to zygotic transition (*Wang et al., 2017*). It also collaborates with FMRP and AGO proteins for translation repression in fly neurons (*Barbee et al., 2006*). In mammals, DDX6 is a general co-factor of the miRNA pathway (*Chen et al., 2014*; *Mathys et al., 2014*; *Kamenska et al., 2016*; *Chu and Rana, 2006*). The intriguing finding of our analysis was that the targets of most tested translation regulators (FRX1-2, FMR1, PUM1-2, most miRNAs...) were AU-rich and had a median behavior in all datasets similar to other mRNAs of same GC content. While the GC bias of the targets of the various RBPs and miRNAs are likely to reflect their sequence preference, some RBPs may not reach sufficient concentration to occupy a significant number of binding sites. Moreover binding does not mean activity, and these factors could require cofactors or post-translational modifications to become 'productive' in terms of mRNA regulation. Of note, while TOP mRNAs clearly constituted an exception in terms of GC content, they are particular too in terms of their translation repression mechanism, with a regulatory sequence located at the 5' end.

PBs add another layer to translation regulation, by storing translationally repressed mRNAs. It was already known that ARE-containing mRNAs bound to ARE-BPs such as TTP and BRF were recruited to PBs (*Franks and Lykke-Andersen, 2007*) and that miRNA targets accumulate in PBs upon miRNA binding in a reversible manner (*Bhattacharyya et al., 2006*; *Liu et al., 2005*). As DDX6 and 4E-T are key factors in PB assembly in mammalian cells (*Minshall et al., 2009*; *Ayache et al., 2015*; *Kamenska et al., 2014*; *Ferraiuolo et al., 2005*), it raises the question of whether these proteins contribute to translation repression by triggering the recruitment of arrested mRNA to PBs, or by mediating translation arrest, which then results in mRNA recruitment to PBs. While we have no answer for DDX6, we observed that 4E-T targets were particularly enriched in PBs, though rather insensitive to DDX6 or PAT1B depletion. First, this suggests that 4E-T function in PBs is partly independent of DDX6, agreeing with the previous observation that some PBs can still form when the DDX6 interaction domain of 4E-T is mutated (*Kamenska et al., 2016*). Second, it indicates that PB localization and translation repression by DDX6 can be separated, at least to some extent.

In addition to their high AU content, we observed that PB mRNAs were longer than mRNAs excluded from PBs. Long CDS could favor mRNA recruitment in PBs by decreasing translation efficiency, and hence increasing the fraction of polysome-free mRNAs. Long 3'UTR should increase the probability of binding translation regulators, contributing also to PB recruitment. In agreement, it is interesting that DDX6 preferentially repressed the translation of mRNAs with long 3'UTR, while the CDS length was irrelevant (*Figure 3—figure supplement 2G*). It is also possible that protein binding over the entire length of mRNA may contribute to PB recruitment. This would explain why the GC content of the 5'UTR has little impact as it is considerably shorter than CDS and 3'UTR. In this

regard, it is interesting to note that we and others have previously proposed from biochemical experiments and electron microscopy imaging that DDX6 and its partner LSM14A coat repressed mRNAs at multiple positions, according to their length (*Ernoult-Lange et al., 2012*; *Götze et al., 2017*).

## Evolutionary issues

Our results raise intriguing issues in terms of evolution. While PBs have been observed in very diverse eukaryotes, animal and vegetal, the GC-rich part of the human genome only emerged in amniotes (the ancestor of birds and mammals) (*Duret et al., 2002*). In more distant organisms, such as yeast, *C. elegans*, *Drosophila* or *Xenopus*, genes have a narrow GC content distribution, most often AU-rich (*Figure 7—figure supplement 1*). Thus, despite the conservation of the DDX6, XRN1 and to a lesser extent PAT1B proteins in eukaryotes, distinct modes of mRNA stability control depending on GC content may have evolved more recently. Moreover, the enzymatic properties of DDX6 could have adapted to the higher GC content of human transcripts.

The GC-rich part of the human genome was acquired through GC-biased gene conversion (gBGC), a non-selective process linked with meiotic recombination affecting GC content evolution (*Duret and Galtier, 2009*). We considered the possibility that meiotic recombination occurred more frequently in genomic regions containing genes involved in basic functions, leading to stronger gBGC and, consequently, to higher GC content of PB-excluded mRNA. However, our analysis showed that mRNA base composition and PB enrichment are associated independently of meiotic recombination or the genomic context. We therefore put forward a model where the genome of higher eukaryotes has evolved partly to facilitate the control of regulators at the translation level, by limiting their protein yield. Regardless, the overall outcome of our study is that in human the GC content, a feature written in the genome, shapes in part mRNA fate and its control in a strikingly coherent system.

## Materials and methods

### Key resources table

| Reagent type (species) or resource | Designation | Source or reference | Identifiers | Additional information |
|---|---|---|---|---|
| Cell line (*Homo sapiens*) | HEK293 | ATCC | Cat# PTA-4488, RRID:CVCL_0045 | |
| Cell line (*Homo sapiens*) | HeLa | ATCC | Cat# CCL-2, RRID:CVCL_0030 | |
| Cell line (*Homo sapiens*) | HCT116 | ATCC | Cat# CCL-247, RRID:CVCL_0291 | |
| Cell line (*Homo sapiens*) | HEK293 expressing GFP-LSM14A | *Hubstenberger et al., 2017* | PMID:28965817 | |
| Transfected construct (rabbit) | siβ-Globin | *Serman et al., 2007* | PMID:17604308 | GGUGAAUGUGGAAGAAGUUdTdT siRNA used as negative control for the siDDX6 expt. |
| Transfected construct (human) | siDDX6 | *Minshall et al., 2009* | PMID:19297524 | GGAACUAUGAAGACUUAAAdTdT |
| Transfected construct (human) | siXRN1 | Thermofisher | Cat# AM16708A | ID125199 |
| Transfected construct (human) | siRNA negative control | Thermofisher | Cat# 4390843 | siRNA used as a negative control for the siXRN1 expt. |
| Transfected construct (human) | shXRN1 | Thermofisher | Cat# RHS 4696–99704634 | Lentiviral 'TRIPZ' construct to transfect and express the XRN1 shRNA. |
| Transfected construct (human) | Non-silencing shRNA | Thermofisher | Cat# RHS 4743 | Lentiviral 'TRIPZ' construct to transfect and express the control shRNA. |

*Continued on next page*

*Continued*

| Reagent type (species) or resource | Designation | Source or reference | Identifiers | Additional information |
|---|---|---|---|---|
| Antibody | anti DDX6 (rabbit polyclonal) | Novus biological | Cat# NB200-191, RRID:AB_10003156 | WB (1:15000), IF (1:2000) |
| Antibody | Anti XRN1 (rabbit polyclonal) | Novus Biosciences | Cat# NB 500–191, RRID:AB_527572 | WB (1:5000) |
| Antibody | Anti XRN1 (rabbit polyclonal) | Bethyl | Cat# A300-443A, RRID:AB_2219047 | WB (1:1000) |
| Antibody | Anti ribosomal S6 (rabbit monoclonal) | Cell signalling technologies | Cat# 2217, RRID:AB_331355 | WB (1:5000) |
| Antibody | Anti Pol II (rabbit polyclonal) | Santa Cruz | Cat# sc-899, RRID:AB_632359 | WB (1:100) |
| Antibody | Anti tubulin (mouse monoclonal) | Sigma-Aldrich | Cat# T9026, RRID:AB_477593 | WB (1:30000) |
| Recombinant DNA reagent | hRluc-GFP-GC-rich | This paper | | phRL-CMV vector bearing an Rluc-GFP GC-rich insert, used in PB-like reconstitution, smiFISH and luciferase reporter expts. |
| Recombinant DNA reagent | Rluc-GFP-AU-rich | This paper | | phRL-CMV vector bearing an Rluc-GFP GC-rich insert, used in PB-like reconstitution, smiFISH and luciferase reporter expts. |
| Sequence-based reagent | ACTB qPCR primers | This paper | | Fwd: TCCCTGGAGAAGAGCTACGA Rev: AGCACTGTGTTGGCGTACAG |
| Sequence-based reagent | APP qPCR primers | Gift from R. Blaise | | Fwd: acttgcatgactacggc Rev: actcttcagtgtcaaagttgt |
| Sequence-based reagent | BACE1 qPCR primers | Gift from R. Blaise | | Fwd: ctttgtggagatggtggac Rev: aaagttactgctgcctgtat |
| Sequence-based reagent | LSM14A qPCR primers | This paper | | Fwd: AGCAGTTTGGTGCTGTTGGT Rev: AACCGCACTACTTTGGGGTA |
| Sequence-based reagent | LSM14B qPCR primers | This paper | | Fwd: CGACAACATCTCTTCTGAACTCAA Rev: GTGTTGAGCTTCCTCTCTTCG |
| Sequence-based reagent | MFN2 qPCR primers | This paper | | Fwd: GAACCTGGAGCAGGAAATTG Rev: AACCAACCGGCTTTATTCCT |
| Sequence-based reagent | PNRC1 qPCR primers | This paper | | Fwd: CCCCCTCAGGAAAGAGGTTTT Rev: ACAAGTGTATACCATGAACAAGCTG |
| Sequence-based reagent | TIMP2 qPCR primers | *Blaise et al., 2012* | PMID:22260497 | Fwd: gaagagcctgaaccacaggt Rev: cggggaggagatgtagcac |
| Sequence-based reagent | TRIB1 qPCR primers | This paper | | Fwd: ACCTGAAGCTTAGGAAGTTCGT Rev: CTGACAAAGCATCATCTTCCCC |
| Sequence-based reagent | HPRT1 qPCR primers | This paper | | Fwd: TAATTGACACTGGCAAAACAATGCAGACT Rev: GGGCATATCCTACAACAAACTTGTCTGGA |
| Sequence-based reagent | REN-lowGC qPCR primers | This paper | | Fwd: CCAGGATTCTTTTCCAATGC Rev: CTTGCGAAAAATGAAGACCTTT |
| Sequence-based reagent | REN-highGC qPCR primers | This paper | | Fwd: CGAGAACGCCGTGATTTT Rev: GACGTGCCTCCACAGGTAG |
| Sequence-based reagent | FIREfly qPCR primers | This paper | | Fwd: TGAGTACTTCGAAATGTCCGTTC Rev: GTATTCAGCCCATATCGTTTCAT |
| Sequence-based reagent | RenGFP-lowGC-24 DNA probe | This paper | | Set of 24 primary probes specific of the RenGFP lowGC mRNA used in smiFISH expts. (See *Supplementary file 2*) |

*Continued*

| Reagent type (species) or resource | Designation | Source or reference | Identifiers | Additional information |
|---|---|---|---|---|
| Sequence-based reagent | RenGFP-highGC-24 DNA probe | This paper | | Set of 24 primary probes specific of the RenGFP highGC mRNA used in smiFISH expts. (see *Supplementary file 2*) |
| Sequence-based reagent | FLAP-Y-Cy3 DNA probe | *Tsanov et al., 2016* | PMID:27599845 | AA TGC ATG TCG ACG AGG TCC GAG TGT AA Secondary probe conjugated to two Cy3 moieties at the 5' and 3' termini. Used in smiFISH expts. |
| Peptide, recombinant protein | CBP-DDX6-HIS | *Ernoult-Lange et al., 2012* | PMID:22836354 | |
| Commercial assay or kit | miRNeasy Mini kit | Qiagen | Cat# 217004 | |
| Commercial assay or kit | TruSeq Stranded Total RNA kit | Illumina | Cat# RS-122–2201 | |
| Commercial assay or kit | Dual-Glo Luciferase assay system | Promega | Cat# E2920 | |
| Chemical compound, drug | Micrococcal Nuclease | Thermo Scientific | Cat# 88216 | |
| Software, algorithm | Cluster 3.0 | http://www.eisenlab.org/eisen/?page_id=42 | RRID:SCR_013505 | |
| Software, algorithm | Java Treeview | https://sourceforge.net/projects/jtreeview/ | RRID:SCR_016916) | |
| Software, algorithm | Morpheus | https://software.broadinstitute.org/morpheus | RRID: SCR_017386 | |
| Software, algorithm | Icy | http://icy.bioimageanalysis.org/ | RRID:SCR_010587 | |
| Software, algorithm | WebGestalt | http://www.webgestalt.org/ | RRID:SCR_006786 | |

## Cell culture and transfection

Human embryonic kidney HEK293 cells, epithelioid carcinoma HeLa cells and colorectal carcinoma HCT116 cells were obtained from ATCC. All cells were tested negative for mycoplasma contamination. HEK293 and HeLa cells were maintained in DMEM supplemented with 10% (v/v) fetal calf serum. HCT116 cells were grown in McCoy's 5A modified medium supplemented with 10% (v/v) fetal bovine serum, 5% (v/v) sodium pyruvate and 5% (v/v) non-essential amino acids. The HEK293 cell line stably expressing GFP-LSM14A (*Hubstenberger et al., 2017*) was maintained under selection using 500 µg/ml Geneticin (Gibco, Life Technology).

For DDX6 silencing, $7.10^5$ cells were transfected at the time of their plating (reverse transfection) with 50 pmoles DDX6 or control β-globin siRNAs (*Minshall et al., 2009*) per 3 cm diameter well, using Lipofectamine 2000 (Life Technologies, France), and split in two 24 hr later. Cells were lyzed 48 hr after transfection.

For XRN1 silencing with siRNAs, $2.10^5$ cells/well were plated in 6-well plates and transfected 24 hr later with 50 nM siRNA negative Control or Silencer Pre-designed siRNA XRN1 (Thermofisher), using Lipofectamine RNAiMAX (Life Technologies). Cells were lyzed 48 hr after transfection.

For XRN1 silencing with shRNA, a doxycycline inducible construct provided by Thermofisher (TRIPZ) with shRNA against XRN1 or non-silencing shRNA was introduced by lentiviral transduction (MOI 0.5). After 10 days of puromycin selection, cells were tested for expression of the construct. For shRNA induction cells were grown to 30% confluency in 10 cm plates before adding 1 µg/ml doxycycline. After 24 hr, cells were split in three and doxycycline was maintained until 48 hr.

For smiFISH experiments and luciferase reporter expression, $2.10^5$ cells were plated in 35 mm diameter dish and transfected 24 hr later with 100 ng of hRluc-GFP-GC-rich or Rluc-GFP-AU-rich plasmids using GenJet Plus DNA (SignaGen Laboratories). For luciferase reporter assay, a Firefly luciferase plasmid (150 ng) was added for normalization. All transfection mixes were made up to 1 µg with pUC19. Cells were processed 24 hr after transfection.

## Construction of DNA plasmids

All plasmids were obtained using the InFusion Advantage PCR cloning kit (Clontech). To obtain medium-size CDS, we cloned the Rluc CDS in frame with the one of GFP. An AU-rich CDS of GFP was amplified from the pUC57-GFP(opt) plasmid (generous gift from N. Campo, *Martin et al., 2010*) and cloned in place of GFP into the BamHI/NotI sites of the pEGFP-N1 plasmid (Clontech) to generate the pGFP(opt)-N1 plasmid. The AU-rich CDS of Rluc was taken from the pRL-TK vector (Promega) and cloned in frame into the pGFP(opt)-N1 plasmid to obtain pRL-GFP(opt) plasmid. The GC-rich CDS of Rluc (hRluc) was taken from the phRL-CMV vector (Promega) and cloned in-frame into the pEGFP-N1 to obtain the phRL-EGFP plasmid. Finally, hRluc-EGFP-GC-rich and Rluc-GFP-AU-rich CDS were cloned into the NheI/NotI sites of the T7 promotor-containing phRL-CMV vector to generate hRluc-GFP-GC-rich and Rluc-GFP-AU-rich plasmids, respectively. The resulting plasmids differ by their AU-rich or GC-rich CDS but encode the same Rluc-GFP fusion protein.

## CBP-DDX6-HIS protein purification

*E.E. coli* BL21-CodonPlus (Novagen) transformed with the CBP-p54-His expression vector (*Ernoult-Lange et al., 2012*) were induced in MagicMedia (Invitrogen) at 16°C during 72 hr to produce the CBP-DDX6-HIS protein. Crude protein extract was prepared and sequentially purified as described previously (*Ernoult-Lange et al., 2012*).

## In vitro transcription and RNA purification

GC-rich and AU-rich RNAs (1707 and 1714 bp, respectively) were transcribed with T7 RNA polymerase (Promega) from linearized (NotI) hRluc-GFP-GC-rich and Rluc-GFP-AU-rich plasmids, purified using the Nucleospin RNA clean-up XS (Macherey-Nagel), quantified with Quantus Fluorometer (Promega) and visualized on 1% agarose gel in 1XTAE buffer containing ethidium bromide.

## Cell-free extract preparation

Stable GFP-LSM14A HEK293 cells grown to 80–90% confluency in 15 cm plates were collected in PBS, cell pellets were frozen in liquid nitrogen and stored at −80°C. Pellets were resuspended in lysis buffer (50 mM Tris, pH 7.4, 1 mM EDTA, 150 mM NaCl, 0,2% Triton X-100) containing 65 U/mL RNaseOut ribonuclease inhibitor (Promega) and EDTA-free protease inhibitor cocktail (Roche Diagnostics), incubated 20 min on ice and centrifuged at 500 xg for 5 min at 4°C to deplete nuclei. The cytoplasmic lysate was half diluted to 75 mM NaCl with buffer containing 50 mM Tris, pH 7.4, 1 mM EDTA, 10 mM CaCl2, 0,2% Triton X-100, and treated with 1000 u/ml of micrococcal nuclease (Thermo Scientific) for 15 min at 37°C. The micrococcal nuclease was inactived by adding EGTA, pH 8.0 to a final concentration of 20 mM. The cytoplasmic lysate was further spun at 11000 xg for 7 min at 4°C to obtain a surpernatant depleted of endogenous P-bodies. This supernatant containing the GFP-LSM14A soluble protein, called cell free extract, was quantified by the Coomassie protein assay (Thermo Scientific).

## In vitro reconstitution assay of P-body like granules

To reconstitute P-body like granules, 1 µg of purified CBP-DDX6-HIS protein was added to 200 µg of cell free extract, mixed or not with RNA, in a 100 µL reaction volume. After 2 hr on ice, 90 µL of the reactions were run through a MACSQuant analyzer (Miltenyi Biotec). Particles were detected according to their Forward-scattered light (FSC) and their green fluorescence using the 488 nm excitation laser and counted in the total volume. We have previously reported that in HeLa cells, 15 µg proteins (corresponding to 53,000 cells) contain 8.6 ng DDX6, and that DDX6 concentration is 0.56 mM in PBs and 3.3 µM in cells (170-fold less) (*Ernoult-Lange et al., 2012*). In the assay, CBP-DDX6-HIS concentration is 0.17 µM (1 µg in 100 µl, 61 kDa), and therefore about 20 fold less concentrated than in a cell, while the cell content (15 µg/53000 cells, 1000 µm$^3$/cell, leading to 200 µg/0.7 µl) is diluted about 100 fold (200 µg in 100 µl).

For imaging experiments, the reactions were centrifuged at 11000 xg for 7 min at 4°C, resuspended in 5 µL of Mowiol (PolySciences) mounting medium, mixed by vortexing and mounted between glass slide and coverslip. Microscopy was performed on a Leica DMR microscope (Leica) using a 63 × 1.32 oil-immersion objective. Photographs were taken using a Micromax CCD camera

(Princeton Scientific Instruments) driven by MetaMorph software (Molecular Devices). Images were processed with NIH ImageJ software.

## smiFISH experiments

Cells transfected for 24 hr with hRluc-GFP-GC-rich or Rluc-GFP-AU-rich plasmids were fixed with 4% paraformaldehyde for 20 min at RT and permeabilized in 70% ethanol overnight at 4°C. The sets of transcript-specific probes (*Supplementary file 2*) and the secondary Cy3 FLAP probe were designed, purchased and hybridized as previously described (*Tsanov et al., 2016*). Of note, none of the AU-rich probes and only 7 out of the 24 GC-rich probes hybridize to the GFP-LSM14A transcripts, resulting in either no signal or a faint signal which was easy to discriminate from the Rluc-GFP mRNA signal. Cells were further processed for immunostaining using rabbit polyclonal anti-DDX6 (1:2000; Novus Biological) and goat anti-rabbit Alexa Fluor 350 (1:300, Thermofisher) antibodies. Epifluorescence microscopy was performed on an inverted Zeiss Z1 microscope equipped with a motorized stage using a 63 × 1.32 oil immersion objective. Images were processed with Icy software.

## Luciferase reporter assay

Cells transfected for 24 hr with Firefly control plasmid and hRluc-GFP-GC-rich or Rluc-GFP-AU-rich plasmids were harvested and processed for RNA (4/5) and protein (1/5) luciferase quantification. Total RNA was purified using Trizol (Invitrogen) and DNAse-treated (Turbo DNAse, Invitrogen). qRT-PCR was carried out as described in the corresponding section, and Renilla mRNA levels were normalized to the Firefly control. Luciferase protein assay was performed with the Dual Glo Luciferase assay kit (Promega) according to the manufacturer's instructions. Relative light determinations were measured in a Lumat LB 9507 luminometer (Berthold).

## Western blot analysis

Total cell lysates were obtained as described previously (*Courel et al., 2006*). Proteins were separated on SDS-PAGE on 4–12% polyacrylamide gel (NuPage, Invitrogen) and transferred onto nitrocellulose membrane (PerkinElmer, France). After blocking in 5% (w/v) nonfat dry milk in PBS for 30 min at RT, the membrane was incubated for 1 hr at 37°C with primary antibodies. After washing in PBS containing 0.05% (v/v) tween-20, blots were incubated for 40 min at RT with horseradish peroxidase-conjugated secondary anti-rabbit antibody (1:10000; Jackson Immunoresearch Laboratories). Immunoreactive bands were visualized by chemiluminescence detection of peroxidase activity (SuperSignal West Pico, Pierce) and exposure to CL-XPosure film (Pierce). Protein expression was evaluated by densitometry (NIH ImageJ).

For the shRNA XRN1 experiment, proteins were isolated using RIPA buffer with Halt Protease and Phosphatase Inhibitor Cocktail (Thermo Scientific), precipitated with acetone and separated on Tris-Acetate 3–8% polyacrylamide gel (NuPage, Invitrogen) before transfer to nitrocellulose membrane (GE Healthcare). After blocking with 5% (w/v) nonfat dry milk in TBST for 30 min at RT the membrane was incubated for 1 hr at RT or o/n at 4°C with primary antibodies. After washing in TBST, blots were subsequently incubated for 1 hr at RT with horseradish peroxidase-conjugated secondary anti-rabbit/mouse antibodies (1:10000; Sigma). Immunoreactive bands were visualized by chemiluminescence detection of peroxidase activity (SuperSignal West Dura, Pierce) and imaged using ImageQuant LAS 4000 (GE Healthcare). Protein expression was evaluated by densitometry (NIH ImageJ).

Primary antibodies were: rabbit polyclonal anti-DDX6 (1:15000; Novus Biological), rabbit polyclonal anti-ribosomal S6 (1:5000; Cell Signaling Technology), rabbit polyclonal anti-XRN1 (1:1000, Bethyl Laboratories, and 1:5000 Novus Bioscience), rabbit polyclonal anti-Pol II (1:100, Santa Cruz), mouse monoclonal anti-tubulin (1:30000, Sigma-Aldrich). *q-RT-PCR analysis* Total RNA (1 µg) was reverse transcribed for 1 hr at 50°C using the SuperScript II First-Strand Synthesis System for RT-PCR (Invitrogen) with 1 µg random primers (Promega). Reverse primers for Firefly and Renilla luciferases were also added for the luciferase reporter assay. No amplification was detected in negative controls omitting the reverse transcriptase. qPCR amplifications (12 µl) were done in duplicates using 1 µl of cDNA and the GoTaq Probe 2X Master Mix (Promega) on a LightCycler 480 (Roche), programmed as follows: 5 min, 95°C; 40 cycles [10 s, 95°C; 15 s, 60°C; 10 s, 72°C]. A last step (5 min progressive

95°C to 72°C) calculating the qPCR product Tm allowed for reaction specificity check. Primers for ACTB, APP, BACE1, LSM14A, LSM14B, MFN2, PNRC1, TIMP2, TRIB1, HPRT1, REN-low-GC, REN-high-GC and FIREfly were either gifts from R. Blaise or designed using the Primer three software (*Untergasser et al., 2012*). The results were normalized using either HPRT1 or FIREfly.

## Library preparation and RNA-Seq data processing

For polysome profiling after DDX6 silencing and transcriptome after PAT1B silencing in HEK293 cells, rRNA was depleted using the Ribo-Zero kit Human/Mouse/Rat (Epicentre), and libraries were prepared using random priming. Triplicate and duplicate libraries were generated from three and two independent experiments, respectively, and processed as detailed previously (*Hubstenberger et al., 2017*; *Vindry et al., 2017*).

For the transcriptome after XRN1 silencing in HeLa cells, libraries were prepared from 500 ng of total RNAs and oligo(dT) primed using TruSeq Stranded Total RNA kit (Illumina) with two technical replicates for each sample. Libraries were then quantified with KAPA Library Quantification kit (Kapa Biosystems) and pooled. 4 nM of this pool were loaded on a high output flowcell and sequenced on a NextSeq500 platform (Illumina) with $2 \times 75$ nt paired-end chemistry.

For the shRNA XRN1 experiment RNA was isolated using Quiazol and miRNeasy Mini Kit (Quiagen), next subjected to DNase treatment (Quiagen) and quality control with Bioanalyzer. The rRNA was removed using rRNA Removal Mix. Libraries were prepared from 1 ug of RNA following TruSeq Stranded Total RNA kit (Illumina) with two technical replicates for each sample. 100nt paired-end RNA-Seq was performed on HiSeq - Rapid Run (Illumina). The results were aligned using hg19 genome and DESeq2, with standard settings, was used for determining FC and p-values.

For PB enrichment, libraries were prepared without prior elimination of rRNA and using random priming. Triplicate libraries were generated from three independent experiments and processed using the same pipeline as for DDX6 silencing (*Hubstenberger et al., 2017*).

For the transcriptome after induction of a stably transfected DDX6 shRNA for 48 hr in K562 cells, the .fastq files from experiments ENCSR119QWQ (DDX6 shRNA) and ENCSR913CAE (control shRNA) were processed according to the same pipeline as DDX6 silencing, except that the control and DDX6 shRNA experiments were not paired to compute the corrected p-values of RNA differential expression in EdgeR_3.6.2.

The ENCODE dataset of mRNAs clipped to DDX6 in K562 cells was generated using ENCODE .bam files aligned on the hg19 genome corresponding to (i) the DDX6 eClip experiment ENCSR893EFU, and (ii) the total RNA-seq of K562 experiment ENCSR109IQO. The enrichment in the CLIP dataset compared to the total RNA sample was calculated as in the DDX6 shRNA experiment.

## Bioinformatic analysis

Briefly, the read coverage was computed as follows. Raw reads were processed using trimmomatics. Alignment was performed on the longest transcript isoforms of Ensembl annotated genes with bowtie2 aligner. Isoforms shorter than 500 nucleotides were not considered. Only unique mapped reads were qualified for counting. Each transcript was subdivided in 20 bins from transcription start site (TSS) to transcript end site (TES) and the proportion of reads for each bin was computed. For the metagene analysis, the average distribution of reads along transcript length was computed so that each gene had the same weight independently of it expression level.

The protein yield was calculated as the ratio between protein abundance in HEK293 cells, taken from *Geiger et al. (2012)*, and mRNA abundance in HEK293 cells, taken from the control sample of our DDX6 polysome profiling experiment. The translation rate was defined as the polysomal to total mRNA ratio, since polysome accumulation can result from both regulated translation and a change in total RNA without altered translation.

For GC profiling of the transcripts in various organisms (*Figure 7—figure supplement 1*), transcripts were downloaded from ENSEMBL (version 92) with their associated gene GC content.

Boxplot representations and statistical tests were performed using the GraphPad Prism software (GraphPad software, Inc) and the R suite (https://www.R-project.org) (R Core Team 2018. R: A language and environment for statistical computing. R Foundation for Statistical Computing, Vienna, Austria). We chose to systematically use the Spearman correlation in the interest of consistency, since the variables under consideration showed both nearly linear (e.g. accumulation in PBs and GC

content of the mRNA) and non-linear (e.g. accumulation in PBs and mRNA length) relationships. Statistical tests for differences between Spearman correlation coefficients were performed using the R package cocor and the 'meng1992' test. Partial correlations were computed using the ppcor package. Other graphical representations were generated using Excel and the Excel Analysis ToolPak (Microsoft). Hierarchical clustering of all transcripts in *Figure 4E* was performed using the Cluster 3.0/Treeview softwares (Kendall's tau distance, average linkage, *de Hoon et al., 2004*). Heatmap representation of the targets of the various regulators in *Figure 5B and D* was performed online using Morpheus (https://software.broadinstitute.org/morpheus).

Gene meiotic recombination rates were computed as crossover rates between gene start and gene end using the genetic map from the HapMap project (*Frazer et al., 2007*). Rates were computed as the weighted average of crossover rates of chromosomal regions that overlap the window.

The enrichment of the GO SLIM categories of cellular component in AU-rich mRNA excluded from PB (Input database: 881 gene IDs, 558 annotated in GO categories) was assessed using the WebGestalt enrichment analysis web tool (*Liao et al., 2019*).

## Datasets used in the bioinformatics analysis

The following datasets were downloaded from the supplementary material of the corresponding papers: 1) For mRNAs containing cis-regulatory motifs: (i) in silico identification: AREs (*Halees et al., 2008*); (ii) experimental determination: CPEs (*Piqué et al., 2008*), G4-containing (*Huppert et al., 2008*) and TOP mRNAs (*Thoreen et al., 2012*). G4-containing genes were restricted to those harboring a G4 in 5'UTR. 2) For RBP targets: (i) PARE in HeLa cells: SMG6 (mRNAs actually cleaved by SMG6) (*Schmidt et al., 2015*) (ii) CLIP in HEK293 cells: HuR and TTP (*Mukherjee et al., 2011*; *Mukherjee et al., 2014*); (iii) CLIP in HeLa cells: YTHDF2 (*Wang et al., 2014*); (iv) RIP-CHIP in HeLa cells: PUM1 (*Galgano et al., 2008*); (v) RIP-CHIP in mouse neurons: 4E-T (*Yang et al., 2014*). In the case of HuR, the transcripts clipped only in 5'UTR/CDS/introns were removed, and the target list was restricted to transcripts clipped more than once. In the case of TTP, the transcripts clipped only in introns were removed.

For other RBP targets (ATXN2, MOV10, IGF2BP1-3, PUM2, FMR1, FXR1-2, AGO1-4, YTHDF2), CLIP data from different laboratories were previously processed through the same pipeline in the CLIPdb 1.0 database using the Piranha method (*Yang et al., 2015*). We retained those performed in epithelial cells (HeLa, HEK293, HEK293T). Moreover, when replicates were available, we selected the RNA-protein interactions detected in at least 50% of the replicates. Except for FXR1-2 targets, we determined whether protein-RNA interactions occurred in UTR or CDS by intersecting coordinates of the read peaks with the v19 gencode annotation, and we removed the transcripts clipped in their CDS.

For miRNA targets, we extracted the list of all experimentally documented targets from miRTarBase (http://mirtarbase.mbc.nctu.edu.tw/php/index.php) (*Hsu et al., 2014*), and selected the targets of the 22 miRNAs of interest.

## Acknowledgements

We thank Marina Pinskaya and Marc Gabriel for scientific discussions and technical assistance. We also thank Virginie Magnone, Kevin Lebrigand (NGS platform, UCA Genomix), Sylvain Baulande, Patricia Legoix-Né, Virginie Raynal (NGS platform, Institut Curie), Nathalie Campo (LMGM, Toulouse) and Régis Blaise (IBPS, Paris).

## Additional information

### Funding

| Funder | Grant reference number | Author |
| --- | --- | --- |
| Association pour la Recherche sur le Cancer | Subvention Fixe | Dominique Weil |
| Agence Nationale de la Recherche | ANR-14-CE09-0013-01 | Dominique Weil |

| European Research Council | DARK consolidator grant | Antonin Morillon |
|---|---|---|
| Agence Nationale de la Recherche | ANR-11-LABX-0028-01 | Antonin Morillon |
| Canceropôle PACA | | Patrick Brest |
| Biotechnology and Biological Sciences Research Council | | Nancy Standart |
| Isaac Newton Trust | | Nancy Standart |
| Fondation Philippe Wiener - Maurice Anspach | | Nancy Standart |

The funders had no role in study design, data collection and interpretation, or the decision to submit the work for publication.

## Author contributions

Maïté Courel, Marianne Bénard, Conceptualization, Formal analysis, Supervision, Validation, Investigation, Visualization, Writing - original draft, Writing - review and editing; Yves Clément, Dominika Foretek, Olivia Vidal Cruchez, Michèle Ernoult-Lange, Conceptualization, Formal analysis, Validation, Investigation, Visualization, Writing - original draft, Writing - review and editing; Clémentine Bossevain, Conceptualization, Formal analysis, Investigation, Visualization, Writing - original draft, Writing - review and editing; Zhou Yi, Conceptualization, Software, Formal analysis, Validation, Investigation, Writing - original draft, Writing - review and editing; Marie-Noëlle Benassy, Formal analysis, Validation, Investigation, Visualization, Writing - original draft, Writing - review and editing; Michel Kress, Conceptualization, Formal analysis, Supervision, Validation, Investigation, Writing - original draft, Writing - review and editing; Caroline Vindry, Conceptualization, Formal analysis, Validation, Investigation, Writing - original draft, Writing - review and editing; Christophe Antoniewski, Conceptualization, Resources, Formal analysis, Writing - original draft, Writing - review and editing; Antonin Morillon, Patrick Brest, Nancy Standart, Conceptualization, Supervision, Funding acquisition, Writing - original draft, Writing - review and editing; Arnaud Hubstenberger, Conceptualization, Supervision, Writing - original draft, Writing - review and editing; Hugues Roest Crollius, Conceptualization, Formal analysis, Supervision, Writing - original draft, Writing - review and editing; Dominique Weil, Conceptualization, Formal analysis, Supervision, Funding acquisition, Validation, Investigation, Visualization, Writing - original draft, Project administration, Writing - review and editing

## Author ORCIDs

Yves Clément  http://orcid.org/0000-0002-5932-9412
Christophe Antoniewski  http://orcid.org/0000-0001-7709-2116
Antonin Morillon  http://orcid.org/0000-0002-0575-5264
Hugues Roest Crollius  http://orcid.org/0000-0002-8209-173X
Dominique Weil  https://orcid.org/0000-0001-7630-1772

## Decision letter and Author response

Decision letter https://doi.org/10.7554/eLife.49708.sa1
Author response https://doi.org/10.7554/eLife.49708.sa2

# Additional files

## Supplementary files

• Supplementary file 1. Transcriptome datasets. Sheet1: polysome profiling after siDDX6 in HEK293 cells. Sheet2: transcriptome after shDDX6 in K562 cells. Sheet3: DDX6 CLIP in K562 cells. Sheet4: transcriptome after siXRN1 in HeLa cells. Sheet5: transcriptome after shXRN1 in HCT116 cells. Sheet6: transcriptome after siPAT1B in HEK293 cells.

• Supplementary file 2. SmiFISH probes sets. Sheet1: AU-rich probes Sheet2: GC-rich probes.

• Transparent reporting form

## Data availability

RNA-Seq gene data have been deposited in SRA under accession codes E-MTAB-4091 for the polysome profiling after DDX6 silencing, E-MTAB-5577 for the transcriptome after PAT1B silencing, and E-MTAB-5477 for the PB transcriptome, all in HEK293 cells. RNA-Seq gene data have been deposited in GEO under accession codes GSE115471 and GSE114605 for the transcriptome after XRN1 silencing in HeLa and HCT116 cells, respectively. ENCODE datasets are available at https://www.encodeproject.org under accession codes ENCSR893EFU for the DDX6 eClip experiment, and ENCSR109IQO for the transcriptome after DDX6 silencing in K562 cells. All data generated or analyzed during this study are included in Supplementary file 1.

The following datasets were generated:

| Author(s) | Year | Dataset title | Dataset URL | Database and Identifier |
|---|---|---|---|---|
| Courel M, Weil D | 2017 | Large-scale study of total and polysomal mRNA after DDX6 depletion in HEK293 cells | https://www.ebi.ac.uk/arrayexpress/experiments/E-MTAB-4091/ | ArrayExpress, E-MTAB-4091 |
| Vindry C | 2017 | RNA-seq of HEK293T cells treated with control b-globin siRNA and Pat1b siRNA | https://www.ebi.ac.uk/arrayexpress/experiments/E-MTAB-5577/ | ArrayExpress, E-MTAB-5577 |
| Hubstenberger A | 2017 | RNA-Seq of purified P-bodies from HEK293 cells | https://www.ebi.ac.uk/arrayexpress/experiments/E-MTAB-5477/ | ArrayExpress, E-MTAB-5477 |

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
