## [Decision Letter]

**Acceptance summary:**

Courel et al. use a variety of genomic datasets to examine the impact of nucleotide content (AU versus GC content) of human mRNAs upon their post-transcriptional fates. The results suggest that translational regulation applies predominantly to AU-rich transcripts, whereas high GC content correlates with mRNA decay as the major mode of regulation for such transcripts. While the cellular mechanism of how the nucleotide content of an mRNA is recognized remains unclear, all three reviewers found the results that are described in this paper interesting and important. The authors have addressed the key concerns during the revision, and the manuscript is now ready for publication.

**Decision letter after peer review:**

Thank you for submitting your article "GC content shapes mRNA storage and decay in human cells" for consideration by *eLife*. Your article has been reviewed by three peer reviewers, and the evaluation has been overseen by a Reviewing Editor and James Manley as the Senior Editor. The reviewers have opted to remain anonymous.

The reviewers have discussed the reviews with one another and the Reviewing Editor has drafted this decision to help you prepare a revised submission.

Summary:

All three reviewers found the results described in this paper interesting and felt that the reported findings are in principle appropriate for publication in *eLife*. However, there were important concerns regarding the statistical methods that must be addressed. We note that the results are mainly correlative and causation was not established. As described below, the authors should test their model with specific reporter mRNAs with varying GC content.

Essential revisions:

1) There is a general issue of how the data are analyzed; almost every result is supported by correlation analysis (Spearman). First and foremost, the authors compare two different correlations many times, and attribute meaning to differences in the Spearman values – this is not appropriate, a statistical test is needed to establish whether two correlations are meaningfully different. Second, the authors use a wide range of adjectives (weak, strong, moderately etc) to describe and interpret correlations, the selection of these adjectives seems rhetorical rather than rigorous. Third, it would be helpful to justify (in the Materials and methods, perhaps) why Spearman rather than Pearson correlations are employed. Fourth, there are statements (e.g., subsections “PBs only accumulate AU‐rich mRNAs”; “The GC content of mRNAs shapes post‐transcriptional regulation”) that require associated statistical tests.

2) Relying on the GC content as the main basis for the conclusion could be problematic for the following reasons: a) There is a clear technical bias that connects GC content to observed transcript abundance as observed through RNAseq. This has been extensively documented (Benjamini et al., NAR 2012, for example), and has to do with the PCR amplification step at the end of almost every library construction protocol. This might not be problem with regards to the data produced by the authors, but they do use data, including CLIP data, from other sources. It can be difficult to be certain that such data is free of such biases. CLIP data may be particularly sensitive to this bias since due to the low amount of input material, a large number of PCR cycles are often required to make a library. b) GC content is not a standalone variable in the transcriptome. It is significantly correlated with a range of other variables, including transcript length (Marin et al. 2003, Yeast), expression (Kudla et al. 2006, PLOS Biology), and conservation (Litterman et al. 2019, Genome Res). Accordingly, all of these variables are correlated with each other. This makes it difficult to assign observed effects to just one of these variables. This is not to say that GC content is not playing a role, perhaps even a dominant one. However, it could be problematic to draw mechanistic conclusions from groups of correlations of metrics (GC, length) that are themselves prone to spurious correlations.

This could be disentangled through the use of reporter transcripts. For example, one could look at the P body localization, DDX6 sensitivity, etc. of long but GC-rich transcripts, short but GC-poor transcripts, and so on. This strategy could also be used to look at the relative contributions of CDS and 3' UTR sequences, since the GC content of the two in endogenous transcripts are correlated and thus also hard to disentangle.

3) The authors should speculate as to how they feel the GC-ness of an mRNA is 'measured' by the cell. Do they feel RNA secondary structure plays a role – in which case the authors could cross-compare with global assessments of RNA structure (e.g. Luu et al. 2016 Cell v165, p1267) Or could it be that the levels of specific amino-acylated tRNAs are critical? In which case the authors could compare to aa-tRNA levels (e.g. Evans et al., Cell v165 p1267). Or is there some other mechanism that the authors feel could be speculated upon?

4) The terminology used throughout the paper needs to more accurately reflect the experiments and data that have been evaluated. For instance- the authors use the terms mRNA stabilization and mRNA decay when describing analyses where steady state mRNA levels have been measured. They silence specific factors and measure steady state mRNA levels and talk about factor-dependent mRNA decay. They haven't at any point in the paper directly measured the rate of mRNA decay or the stability of an mRNA. In fact, the only experiment that touches upon this is the metaplot of reads across RNAs showing accumulations of 5' and 3' reads. Again, though this is not a direct measure. Another example of this is the assumption that polysome enrichment equates to translation rates. Once again translation or the rate of translation has never been measured in the paper. Instead the steady state levels of mRNA in polysome fractions has been measured and is used as a proxy for translation rates. While this is quite common in the general literature, at the very least the authors needs to explain the assumption and the potential for polysome enrichment due to inhibited translation elongation should be raised as a potential caveat.

5) Elements in the Discussion and the legend titles to individual parts of figures are often overstated and need to be looked at carefully – several examples are found throughout the text.

---

## [Author Response]

Essential revisions:1) There is a general issue of how the data are analyzed; almost every result is supported by correlation analysis (Spearman). First and foremost, the authors compare two different correlations many times, and attribute meaning to differences in the Spearman values – this is not appropriate, a statistical test is needed to establish whether two correlations are meaningfully different. Second, the authors use a wide range of adjectives (weak, strong, moderately etc) to describe and interpret correlations, the selection of these adjectives seems rhetorical rather than rigorous. Third, it would be helpful to justify (in the Materials and methods, perhaps) why Spearman rather than Pearson correlations are employed. Fourth, there are statements (e.g., subsections “PBs only accumulate AU‐rich mRNAs”; “The GC content of mRNAs shapes post‐transcriptional regulation”) that require associated statistical tests.

(i) Concerning the comparison between Spearman correlation coefficients, we have now applied a statistical test to confirm that the discussed differences were meaningful. The p-values have been introduced in the Results section and the R package used is indicated in the Materials and methods.

(ii) We have now unified the description of correlations using the following rule: Rs ≤ │0.25│: weak; │0.25│ < Rs < │0.5│: moderate; Rs ≥│0.5│: strong.

(iii) The Pearson correlation evaluates the linear relationship between two variables, whereas the Spearman correlation can also evaluate variables with a non-linear relationship. We chose to systematically use the Spearman correlation in the interest of consistency, since the variables under consideration showed both nearly linear (e.g. accumulation in PBs and GC content of the mRNA, new Figure 1—figure supplement 1D) and non-linear (e.g. accumulation in PBs and mRNA length, new Figure 1—figure supplement 1A) relationship. This justification has been added to Materials and methods.

(iv) We have now added the missing information. For the statement that PB mRNAs are shorter than SG mRNAs, a new figure panel (Figure 1—figure supplement 1C) provides a full description of the distribution of length of mRNAs enriched and excluded from PBs and SGs, with the requested statistical test. We apologize for initially omitting the statistical test supporting the difference of GHI in PB-enriched versus PB-excluded mRNAs. It is now indicated in Figure 6C legend.

2) Relying on the GC content as the main basis for the conclusion could be problematic for the following reasons: a) There is a clear technical bias that connects GC content to observed transcript abundance as observed through RNAseq. This has been extensively documented (Benjamini et al., NAR 2012, for example), and has to do with the PCR amplification step at the end of almost every library construction protocol. This might not be problem with regards to the data produced by the authors, but they do use data, including CLIP data, from other sources. It can be difficult to be certain that such data is free of such biases. CLIP data may be particularly sensitive to this bias since due to the low amount of input material, a large number of PCR cycles are often required to make a library.

While we are aware that some GC bias can be introduced during library preparation, we do not believe that they are responsible for our results. Concerning our own datasets (PB transcriptome, transcriptomes after silencing), the extent of the GC bias was very strong, and highly similar when analyzing separately the replicates of each experiment, as well as in silencing experiments performed in parallel in two laboratories using different protocols and cell lines. For the datasets from other sources, note that some of them do not rely on RNAseq (ARE-, G4- and CPE- containing mRNAs) or only partially (experimentally validated miR targets). For the others, we agree that some artefactual GC bias cannot be completely excluded. Consequently, whenever possible, we repeated our analysis using independent target lists (for YTHDF2), or using the targets identified in several CLIP experiments (for IGFBP1, FMR1, ATXN2, AGO1, AGO2). Regardless of origin or approach, overall, the observed GC biases were highly consistent: (i) SMG6 targets defined by PARE were GC rich, as observed for NMD targets using different approaches (Gingold et al., 2014, see Discussion); (ii) both the targets of ARE-binding proteins (HuR and TTP) and the ARE-containing mRNAs were AU-rich; (iii) same for miRNA and AGO targets; (iv) or for PUM1 and PUM2 targets, which are known to overlap. Therefore, we do not see a strong argument to reconsider our global conclusions.

b) GC content is not a standalone variable in the transcriptome. It is significantly correlated with a range of other variables, including transcript length (Marin et al. 2003, Yeast), expression (Kudla et al. 2006, PLOS Biology), and conservation (Litterman et al. 2019, Genome Res). Accordingly, all of these variables are correlated with each other. This makes it difficult to assign observed effects to just one of these variables. This is not to say that GC content is not playing a role, perhaps even a dominant one. However, it could be problematic to draw mechanistic conclusions from groups of correlations of metrics (GC, length) that are themselves prone to spurious correlations.This could be disentangled through the use of reporter transcripts. For example, one could look at the P body localization, DDX6 sensitivity, etc. of long but GC-rich transcripts, short but GC-poor transcripts, and so on. This strategy could also be used to look at the relative contributions of CDS and 3' UTR sequences, since the GC content of the two in endogenous transcripts are correlated and thus also hard to disentangle.

We fully agree that GC content correlates with a variety of other variables. Throughout the manuscript, we tried to tackle these potential confounding effects: GC content at the transcript and genomic levels (first section of the Results), GC content and length of the mRNA (Figure 1E, Figure 1—figure supplement 1H), GC content and length of the CDS (Figure 2E), GC content of the CDS and 3’UTR (Figure 6A, B). In principle, the methodology we used enables to disentangle parameters two by two, using our experimental dataset. For instance, take GC content and length of the transcript (Figure 1E): considering mRNAs shorter than 1.5 kb, those which are out of PBs have a higher GC content than the ones which are enriched in PBs (56% vs. 46% median GC); and the same is true in all length ranges. Therefore, even if GC content and length are correlated, we can conclude that PB mRNAs have a lower GC content independently of their length.

Concerning the possible confounding effect of expression level and conservation, we have now added in Figure 1—figure supplement 1G a new analysis relying on the computing of partial correlations. It shows that the correlation between GC content and PB localization does not result from the correlation between GC content and expression levels or conservation.

The reviewers suggested we engineer reporter genes with combinations of particular features. Putting aside the numerous combinations to test if we are to consider the length and GC content in 5’UTR, CDS and 3’UTR, meeting the objective would require to have a clearer knowledge of all important parameters, while we are far from that. For instance, a number of RBPs, or combination of them, are likely to play a role in PB localization. How to integrate such a complexity in synthetic reporters? Moreover, RBP binding sites often follow broad consensus, so that by manipulating the GC content of the reporter, one may unintentionally create RBP binding sites, which could also lead to confounding effects.

Nevertheless, following the reviewers’ suggestion, we have now added a reporter assay to test one of the simplest issue : can the GC content of the CDS affect PB localization ? We expressed in cells a GC-rich or AU-rich luciferase mRNA and monitored its accumulation in PBs using single molecule FISH and a PB marker. Clusters of AU-rich mRNAs were found five-times more often in PBs than clusters of GC-rich mRNAs. This was observed in a HEK293 cell clone expressing the PB protein GFP-LSM14A, as well as in parental HEK293 cells using DDX6 immunostaining. These new data are presented in Figures 6D-F and Figure 6—figure supplement 2A-B.

3) The authors should speculate as to how they feel the GC-ness of an mRNA is 'measured' by the cell. Do they feel RNA secondary structure plays a role – in which case the authors could cross-compare with global assessments of RNA structure (e.g. Luu et al. 2016 Cell v165, p1267) Or could it be that the levels of specific amino-acylated tRNAs are critical? In which case the authors could compare to aa-tRNA levels (e.g. Evans et al., Cell v165 p1267). Or is there some other mechanism that the authors feel could be speculated upon?

We agree with the reviewers that the mechanism by which the GC content plays a role is a burning issue. Investigating the potential role of RNA secondary structures will require detailed descriptors and in depth analysis, since we know that the global RNA secondary structure correlates with the GC content. This seemed to us out of reach for the current manuscript, but we now mention this possibility at the end of the Results section.

However, we have now set-up a cell-free assay to investigate the capacity of AU-rich and GC-rich RNA to enhance the formation of PB-like granules. In this assay, granules form upon addition of DDX6 protein to an ice-cold cytoplasmic extract containing the PB marker LSM14A fused to GFP. The granules are then counted by flow cytometry, an approach that we previously used for PB purification (we used FAPS, Fluorescence Activated Particle Sorting) (Hubstenberger et al., Mol Cell 2017). The addition of synthetic AU-rich RNA but not GC-rich RNA favors the formation of DDX6-dependent granules that contain LSM14A. This is in support of the GC content playing, at least in part, a translation-independent role in PB assembly. Further studies will be required to define whether it is important for the binding of particular RBPs and/or for the structure of the RNA. These new data are presented in Figures 6G, H and Figure 6—figure supplement 2C-F.

Concerning the effect of the GC content on codon usage, we followed the suggestion to cross our data with the abundance of amino-acylated tRNAs. There has been active discussion on the extent of adaptation of codon usage to tRNA abundance in eukaryotes, particularly in human. Despite their high protein yield, the codon usage in PB-excluded mRNAs is not clearly fitted to the abundance of the isoacceptors. Other mechanisms such as tRNA modifications or differences of tRNA affinity for their codon may be more suited to understand the consequences of codon usage bias. While this is a negative result, we still added the analysis in Figure 2—figure supplement 2, as it could be of interest for other readers.

4) The terminology used throughout the paper needs to more accurately reflect the experiments and data that have been evaluated. For instance- the authors use the terms mRNA stabilization and mRNA decay when describing analyses where steady state mRNA levels have been measured. They silence specific factors and measure steady state mRNA levels and talk about factor-dependent mRNA decay. They haven't at any point in the paper directly measured the rate of mRNA decay or the stability of an mRNA. In fact, the only experiment that touches upon this is the metaplot of reads across RNAs showing accumulations of 5' and 3' reads. Again, though this is not a direct measure. Another example of this is the assumption that polysome enrichment equates to translation rates. Once again translation or the rate of translation has never been measured in the paper. Instead the steady state levels of mRNA in polysome fractions has been measured and is used as a proxy for translation rates. While this is quite common in the general literature, at the very least the authors needs to explain the assumption and the potential for polysome enrichment due to inhibited translation elongation should be raised as a potential caveat.

Concerning mRNA decay, we acknowledge that we did not measure any decay rate throughout the study, but only analyzed steady-state levels following the silencing of XRN1, DDX6 and PAT1B. XRN1 is a 5’ exonuclease, and DDX6 and PAT1B have well-evidenced roles in mRNA decay and translation, but not in transcription. We therefore feel justified in assuming that the global changes in mRNA levels that we are documenting mostly result from changes in mRNA decay or stabilization, rather than in transcription. To take into account the reviewer’s comment, we modified the text to reflect more accurately the experiments. However, it was impossible to do it systematically without becoming unreadable. Therefore, we extended our previous comment “Since DDX6 is cytoplasmic [Ernoult-Lange et al., 2009], we assumed that total mRNA accumulation generally reflected their increased stability” to raise the potential caveat: “though we cannot exclude altered transcription levels for some of them.” We provide similar arguments for PAT1B data.

Concerning translation, we agree that we did not measure the rate of translation. Instead, as is common in the field, we used the ratio of polysome to total mRNA levels as a proxy of translation rate, as explained: “As polysomal accumulation can result from both regulated translation and a change in total RNA without altered translation, we then used the polysomal to total mRNA ratio as a proxy measurement of translation rate.” Following the reviewer’s comment, we extended this sentence to raise the potential caveat of relying on this ratio: “Nevertheless, for few transcripts, polysomal enrichment may reflect an elongation block rather than an increased rate of initiation”. We also recall in Materials and methods and legends what was really measured.

5) Elements in the Discussion and the legend titles to individual parts of figures are often overstated and need to be looked at carefully – several examples are found throughout the text.

We have modified the manuscript to avoid overstatements.